Insight on the anatomy, systematic relationships, and age of the Early Cretaceous ankylopollexian dinosaur Dakotadon lakotaensis

Boyd Clint A. 1 clintboyd@stratfit.org
Pagnac Darrin C. 2
1 North Dakota Geological Survey , Bismarck, North Dakota , USA
2 Department of Geology and Geological Engineering, South Dakota School of Mines and Technology , Rapid City, South Dakota , USA
Farke Andrew
Electronic publication date: 2015 Sep 22
Publication date: 2015
Volume: 3
Electronic Location ID: e1263
Received 2015 May 28; Accepted 2015 Sep 3
Copyright: © 2015 Boyd and Pagnac
Copyright year: 2015
Copyright holder: Boyd and Pagnac
License: This is an open access article distributed under the terms of the Creative Commons Attribution License, which permits unrestricted use, distribution, reproduction and adaptation in any medium and for any purpose provided that it is properly attributed. For attribution, the original author(s), title, publication source (PeerJ) and either DOI or URL of the article must be cited.
License URL: https://creativecommons.org/licenses/by/4.0/

Keywords: Dakotadon, Lakota Formation, Early Cretaceous, Ornithischia, Ankylopollexia, South Dakota, Bayesian, Systematics

Funding: South Dakota School of Mines & Technology This research was funded via the Haslem Postdoctoral Fellowship at the South Dakota School of Mines & Technology. The funders had no role in study design, data collection and analysis, decision to publish, or preparation of the manuscript.

==============================
Knowledge regarding the early evolution within the dinosaurian clade Ankylopollexia drastically increased over the past two decades, in part because of an increase in described taxa from the Early Cretaceous of North America. These advances motivated the recent completion of extensive preparation and conservation work on the holotype and only known specimen of Dakotadon lakotaensis, a basal ankylopollexian from the Lakota Formation of South Dakota. That specimen (SDSM 8656) preserves a partial skull, lower jaws, a single dorsal vertebra, and two caudal vertebrae. That new preparation work exposed several bones not included in the original description and revealed that other bones were previously misidentified. The presence of extensive deformation in areas of the skull is also noted that influenced inaccuracies in prior descriptions and reconstructions of this taxon. In addition to providing an extensive re-description of D. lakotaensis, this study reviews previously proposed diagnoses for this taxon, identifies two autapomorphies, and provides an extensive differential diagnosis. Dakotadon lakotaensis is distinct from the only other ankylopollexian taxon known from the Lakota Formation, Osmakasaurus depressus, in the presence of two prominent, anteroposteriorly oriented ridges on the ventral surfaces of the caudal vertebrae, the only overlapping material preserved between these taxa. The systematic relationships of D. lakotaensis are evaluated using both the parsimony and posterior probability optimality criteria, with both sets of analyses recovering D. lakotaensis as a non-hadrosauriform ankylopollexian that is more closely related to taxa from the Early Cretaceous (e.g., Iguanacolossus, Hippodraco, and Theiophytalia) than to more basally situated taxa from the Jurassic (e.g., Camptosaurus, Uteodon). This taxonomic work is supplemented by field work that relocated the type locality, confirming its provenance from unit L2 (lower Fuson Member equivalent) of the Lakota Formation. Those data, combined with recently revised ages for the members of the Lakota Formation based on charophyte and ostracod biostratigraphy, constrain the age of this taxon to the late Valanginian to early Barremian.

Introduction

Knowledge of basal iguanodontian and ankylopollexian dinosaurs from the early Cretaceous of North American has improved considerably in recent decades. As a result of sustained surveys of Lower Cretaceous strata, several new taxa were recognized from Texas (Winkler, Murry & Jacobs, 1997), Colorado (Foster, 2003; Brill & Carpenter, 2007), and Utah (DiCroce & Carpenter, 2001; Gilpin, DiCroce & Carpenter, 2007; Carpenter & Wilson, 2008; McDonald et al., 2010) (Fig. 1). Additionally, thorough reviews of previously named taxa were conducted, clarifying diversity and distribution (e.g., Carpenter & Wilson, 2008; Paul, 2008; Carpenter & Ishida, 2010; McDonald, 2011). Despite this progress, recent attempts to resolve the systematic relationships of these taxa (McDonald, Barrett & Chapman, 2010; McDonald, 2011) were impeded as many taxa are based on highly fragmentary material, much of which is either too incomplete to include in a comprehensive analyses or which preserves portions of the skeleton largely unknown in other taxa.

Figure 1 Geographic distribution of Early Cretaceous iguanodontians in North America.

Taxa found at each locality are as follows: (A) Dakotadon lakotaensis; (B) Osmakasaurus depressus; (C) Tenontosaurus tilletti, Tenontosaurus dossi; (D) Tenontosaurus tilletti; (E) Theiophytalia kerri; (F) Cedrorestes crichtoni, Planicoxa venenica; (G) Hippodraco scutodens; (H) Iguanacolossus fortis; (I) Tenontosaurus sp; (J) Tenontosaurus tilletti.

Lower Cretaceous terrestrial strata of the Black Hills region of South Dakota yield a modest flora and fauna, but a lack of dedicated paleontological surveys results in a limited understanding of the paleontology of these units. The Lakota Formation of the Inyan Kara Group (Fig. 1B) is primarily known for its flora, most notably abundant petrified wood and Cycadeoides (Ward, 1899; Weiland, 1916). Trace fossils of both vertebrates and invertebrates are common in the Lakota Formation (Anderson, 1973; Lockley, Janke & Theisen, 2001; Way et al., 1998), and vertebrate occurrences include Chondrichthyes (Cicimurri, 1998), Osteichthyes (Martin & Rich, 1987); Testudinata (Martin & Rich, 1987), and triconodont and dryolestid mammals (Cifelli, Davis & Sames, 2014). Limited dinosaurian material has also accumulated from the Lakota Formation for more than a century. That material is often isolated and fragmentary, but has generally resulted in the description of new taxa given the relatively sparse record of dinosaurian remains from this interval in North America. Notable dinosaur occurrences from the Lakota Formation include Osmakasaurus (=Camptosaurus) depressus from Calico Canyon (Fig. 1B) (Gilmore, 1909; McDonald, 2011), the ankylosaurian Hoplitosaurus (=Stegosaurus) marshi (Lucas, 1901; Lucas, 1902); an unidentified neosauropod (D’Emic & Foster, 2014), and an isolated femur referred to “Hypsilophodon” weilandi (Galton & Jensen, 1979), the latter of which is now considered a nomen dubium (Galton, 2012).

In November of 1975, Dale Rossow brought a partial skull and associated postcrania from the Lakota Formation that was collected by his father, Louis Rossow, to the attention of South Dakota School of Mines and Technology professor emeritus John Willard, who in turn brought it to Philip Bjork of the Museum of Geology. Louis Rossow collected geologic specimens from outcrops on family homesteads in Whitewood Valley. After discovering this specimen, Louis assembled a crew of family members to carve the material from a small outcrop of the Lower Cretaceous Lakota Formation. That specimen, SDSM 8656, was subsequently donated to the museum and described as the holotype of Iguanodon lakotaensis, which was considered at that time to represent the earliest record of that genus in North America (Weishampel & Bjork, 1989). The disparity between SDSM 8656 and Iguanodon was eventually recognized and a new genus, Dakotadon, was erected for that species (Paul, 2008). Dakotadon lakotaensis remains the most complete Early Cretaceous dinosaur from the Black Hills region, with SDSM 8656 as the only described specimen.

Given that SDSM 8656 was donated to the museum after collection, the original work by Weishampel & Bjork (1989) lacked detailed stratigraphic information, resulting in substantial uncertainty as to the age of D. lakotaensis. Thus, it was deemed necessary to attempt to place SDSM 8656 in a more refined stratigraphic and temporal context. In the spring of 2014, the authors returned to the site of discovery of Dakotadon lakotaensis east of Whitewood, South Dakota guided by Russell and LaVon Yuill, the grandchildren of Louis Rossow. The original locality was located (Fig. 2) and detailed stratigraphic information was recorded (Fig. 3). Concurrently, during the spring of 2014 extensive conservation was devoted to SDSM 8656. Throughout the course of those efforts, several new features were revealed that were not apparent during the original description and several fragments that were entirely encased in sediment were exposed and connected with the rest of the specimen, providing important anatomical information regarding this species. This study details the results of those efforts, providing a full redescription of SDSM 8656, increased resolution of the stratigraphic position and approximate age of D. lakotaensis, and a reassessment of the systematic relationships of this species within Ankylopollexia.

Figure 2 Type locality of Dakotadon lakotaensis.

Main photograph (facing north) shows an overview of the hillside containing the outcrop from which SDSM 8656 was collected (red arrow indicates precise location). Darrin Pagnac is standing at the base of the Lakota Formation outcrop in this area, and the stratigraphic section presented in Fig. 3 runs from his feet to the top of the hill in the upper right corner of the photograph. The inset photograph in the upper left corner (facing east) shows where SDSM 8656 was removed (dashed lines).

Figure 3 Stratigraphic section recorded at the type locality of Dakotadon lakotaensis.

Stratigraphic column in the middle, with unit descriptions along the right side and close up images of select intervals along the left side. Total exposed section is just under fifteen meters. The lower contact with the underlying Morrison Formation is not exposed, although isolated outcrops of Morrison Formation are present nearby on the same hillside. Position of SDSM 8656 is interpreted to be within unit L2, which correlates with the lower Fuson Member. Abbreviations: c, coarse sandstone; cl, clay; f, fine sandstone; ft, feet; M, meters; m, medium sandstone; s, silt; vf, very fine sandstone.

Materials & Methods

Specimen preparation methods

An extensive array of preparation techniques were used to repair SDSM 8656 and to complete finish preparation of the specimen, which was not done prior to the original description. Many areas of the specimen had suffered breaks owing to failed glue joints and the entire specimen was coated in polyvinyl acetate. There were also isolated areas where it is suspected cyanoacrylates were applied. Previous preparation work also included infilling missing portions of the specimen with plaster of paris and wood putty to add stability, and the insertion of a series of wire rods into the bone to reattach the anterior portions of the premaxillae to the rest of the skull.

A solution of 50% acetone and 50% ethanol (weight to weight solution) was applied to remove the coating of previously applied adhesives on the external surfaces. The plaster and wood putty were removed manually using a Paleotools Micro Jack #1, and the wire rods were removed once those supporting materials were excised. Any remaining adhesive and filler material was removed using micro air abrasion on a Comco MB1000 using sodium bicarbonate powder.

Finish preparation of SDSM 8656 took the longest amount of time to complete, as some bones were still completely encased in sediment and were never included in the original description. Large patches of matrix were removed using a Paleotools Micro Jack #1 until the bone surface was approached. The remaining surficial matrix was removed with micro air abrasion as described above. Broken pieces that could be solidly reattached were glued using ethyl methacrylate co-polymer Paraloid® B72 in acetone (30% weight to weight solution). Large gaps in the specimen were filled with a mixture of finely ground matrix previously removed from the specimen and Paraloid® B72 in acetone (30% weight to weight solution). Once preparation was complete, the entire specimen was lightly coated using Paraloid® B72 in ethanol (5% weight to weight solution) to ensure surface stability. Once work was completed, the specimen was returned to collections and stored in a custom fit cavity mount constructed using ethafoam and Tyvek®.

Geological field methods

Field work was conducted in April and November of 2014; initial assessment of the site was conducted first and a detailed stratigraphic section was compiled in November. A single stratigraphic section was measured at the type locality. Outcrop section was measured with a Brunton compass and Jacob staff as described by Compton (1985). The sedimentologic characteristics of each unit were assessed visually or through comparison with standard grain size charts. Overall lithostratigraphic correlations were made with additional outcrops of Lakota Formation near the Dakotadon lakotaensis type locality, and south of the town of Sturgis, SD. Assignment to lithostratigraphic members or informal subunits of the Lakota Formation was based on those comparisons.

Geologic Setting

SDSM 8656 was recovered from the Lakota Formation, the most prominent lower Cretaceous nonmarine unit in the Black Hills, although Weishampel & Bjork (1989) did not provide a precise stratigraphic position or even identify a specific member of the Lakota Formation. Exposures of Lakota Formation are found on the periphery of the Black Hills in both South Dakota and Wyoming (Fig. 1). Throughout much of the region, the Lakota Formation overlies the interfingering units of the upper Jurassic Unkpapa and Morrison formations, and is overlain by the transgressive marine beds of the Fall River Formation (Waage, 1959). Together, the Lakota and Fall River formations comprise the Inyan Kara Group in South Dakota. Three members of the Lakota Formation are recognized, the basal Chilson Member, the Minnewaste Limestone Member, and the upper-most Fuson Member (Darton, 1901; Rubey, 1931; Waage, 1959; Post & Bell, 1961). The Lakota Formation is interpreted to be temporally equivalent to portions of both the Cloverly Formation of Wyoming and Montana, and the Cedar Mountain Formation of eastern Utah (Way et al., 1998; Zaleha, 2006; Sames, Cifelli & Schudack, 2010; Martin-Closas, Sames & Schudack, 2013; Cifelli, Davis & Sames, 2014).

Most detailed descriptions of the Lakota Formation are focused on the extensive deposits present in the southern Black Hills (Dahlstrom & Fox, 1995), or on distinctive beds in northeastern Wyoming (Way et al., 1998). Each member is inconsistently represented in outcrop throughout the Black Hills. Exposures from the south record a thicker and much more complete section, whereas those from the east and north are often truncated and missing thick intervals, including the complete absence of the Minnewaste Limestone (Dahlstrom & Fox, 1995; Way et al., 1998). Correlation throughout the region is notably difficult, as is defining precise ages. The tripartite division of the Lakota Formation proposed by Way et al. (1998), including units L1, L2, and L3, focused specifically on exposures in the northern and eastern Black Hills. Unit L1, corresponding to the Chilson Member of other workers, is recognized by a predominance of interbedded siltstone and mudstone, and numerous coal beds. At outcrops near Sturgis, SD, this unit forms an angular unconformity with L2. Unit L2, corresponding to the lower Fuson Member, is characterized by cliff-forming quartz sandstone beds with numerous angular gray and white claystone chips. Unit L3, corresponding to the upper Fuson Member, consists of fine-grained quartz arenites interbedded with white and red mudstone. In the latter unit, cobble- to boulder-sized clasts can be found at the base of some beds, and Arenicolites burrows are common.

The exposed section at the type locality (SDSM V 2015-1: Figs. 2 and 3) encompasses approximately 15.0 m of resistant sandstone (Fig. 3). Four distinct stratigraphic units are recognized (Fig. 3), with SDSM 8656 recovered from the lower-most unit. The observed sequence is typical of the Lakota Formation, consisting predominantly of medium to fine grained, buff to rust colored sandstone. Distinctive features include prominent, elongate iron nodules low in the section, and multiple, 4.0–6.0 mm thick mud draped stringers that are abundant in upper units. The section is capped by a distinctive red, ripple marked sandstone, which is more indurated than the lower units (Fig. 3). A 2.0–3.0 mm thick gray, silty clay demarcates the contact between the upper-most two units.

Our interpretation of the section at the type locality of D. lakotaensis best matches the description of unit L2, primarily based on the presence of graded sandstone beds with numerous angular clay clasts and the observation of mud-draped stringers near the top of the section. For comparison, we located an angularly unconformable contact between units L1 and L2 south of Sturgis, SD. Units above that contact matched those from the D. lakotaensis type locality, containing similar, cliff-forming sandstone beds, iron nodules, angular clay clasts, and mudstone stringers. The absence of boulder-sized clasts and burrows at the type locality precludes referral of that section to unit L3. Based on these interpretations, the type locality is situated within unit L2, which is equivalent to the lower portion of the Fuson Member. These findings contrast with the statement by Carpenter & Ishida (2010) that reported the horizon of the type specimen of Dakotadon lakotaensis as the Chilson Member of the Lakota Formation, although no supporting information was given for that referral. Most other prior reports only specified the Lakota Formation, without identifying a specific member (e.g., Weishampel & Bjork, 1989; Paul, 2008).

Early interpretations of the age of the Lakota Formation varied from Valanginian to Aptian (Sohn, 1958; Sohn, 1979; Anderson, 1973; Cook & Bally, 1975). Recent interpretations based on ostracod biostratigraphy (Sames, Cifelli & Schudack, 2010), charophytes (Martin-Closas, Sames & Schudack, 2013), and mammalian biochronology (Cifelli, Davis & Sames, 2014) extend the lower-most units to the Berriasian and limit the upper-most to the Barremian. Our interpretation of the stratigraphic position of the type locality, in the lower Fuson Member, suggests a late Valanginian to early Barremian age (Cifelli, Davis & Sames, 2014). This is slightly older than the Barremian (Weishampel & Bjork, 1989; DiCroce & Carpenter, 2001; Norman, 2004; Paul, 2008; You & Li, 2009) or Aptian (Norman, 1998) ages previously reported for D. lakotaensis, although Carpenter & Ishida (2010) did assign a Valanginian age to this taxon.

Systematic Paleontology

DINOSAURIA Owen, 1842	
ORNITHISCHIA Seeley, 1887	
ORNITHOPODA Marsh, 1881 (sensu Butler, Upchurch & Norman, 2008)	
ANKYLOPOLLEXIA Sereno, 1986 (sensu Sereno, 2005)	
DAKOTADON Paul, 2008	

Name bearing species:Dakotadon lakotaensis (Weishampel & Bjork, 1989)

Other included species: None.

Diagnosis: As for type and only known species.

DAKOTADON LAKOTAENSIS (Weishampel & Bjork, 1989)

Figs. 4–9 and 11

Iguanodon lakotaensisWeishampel & Bjork, 1989:57 Figs. 1–7

“Iguanodon” lakotaensis Brill & Carpenter, 2007:53 Fig. 3.6B.

cf. Iguanodon lakotaensis Norman, 2015:150

Figure 4 Anterior portion of the skull of Dakotadon lakotaensis (SDSM 8656).

(A) photograph in left lateral view; (B) photograph in right lateral view; (C) photograph in ventral view. Abbreviations: ant, anterior; post, posterior; vent, ventral. Scale bars equal 5.0 cm.

Figure 5 Close ups of the anterior portion of the skull of Dakotadon lakotaensis (SDSM 8656).

(A) photograph in left lateral view of the antorbital fenestra and surrounding bones; (B) photograph in ventral view of the premaxillae and anterior portions of the maxillae. In (A), the white dashed line follows the contact between the lacrimal and prefrontal and the black dashed lines outline the anterior process of the jugal along its contacts with the lacrimal (dorsal) and maxilla (ventral). Abbreviations: af, antorbital fossa; ant, anterior; dam, damaged area; eaof, external antorbital fenestra; ju, jugal; la, lacrimal; lat, lateral; mx, maxilla; na, nasal; pavf, premaxillary anteroventral foramen; pd, premaxillary denticle; pf, prefrontal; plp, posterolateral process of premaxilla; pm, premaxilla; pmf, posterior maxillary foramen; post, posterior; ppvf, premaxillary posteroventral foramen; vent, ventral; vo, vomer. Scale bars equal 4.0 cm.

Figure 6 Left side of the skull of Dakotadon lakotaensis (SDSM 8656) in dorsal and slightly medial view.

Abbreviations: ant, anterior; dam, damaged area; eaof, external antorbital foramen; ep, ectopterygoid; ju, jugal; la, lacrimal; mx, maxilla; na, nasal; pal, palatine; pf, prefrontal; pmf, posterior maxillary foramen; ppf, postpalatine foramen; vent, ventral; vo, vomer. Scale bars equal 5.0 cm.

Figure 7 Anterior portion of the braincase and skull roof of Dakotadon lakotaensis (SDSM 8656).

(A) Photograph in dorsal view showing preserved portions of the frontals, posterior portion of the left prefrontal, left palpebral, left postorbital, anterior portion of left squamosal, and anterior portion of parietals; (B) photograph in ventral view showing preserved portions of the frontals, posterior portion of the left prefrontal, left palpebral, left postorbital, anterior portion of left squamosal, anterior portion of parietals, anterodorsal portions of the right and left laterosphenoids, and dorsal portion of the orbitosphenoid; (C) photograph in left lateral view showing preserved portions of the frontals, posterior portion of the left prefrontal, left palpebral, part of the left postorbital (posterior-most portion not shown), anterodorsal portions of the right and left laterosphenoids, and dorsal portion of the orbitosphenoid. Abbreviations: fr, frontal; lat, lateral; ls, laterosphenoid; os, orbitosphenoid; ot, optic tract; par, parietal; pf, prefrontal; po, postorbital; post, posterior; sa, supraorbital articulation surfaces; sor, supraorbital; vent, ventral. Scale bars equal 5.0 cm.

Holotype: SDSM 8656: Partial skull, lower jaws, and associated dorsal and caudal vertebrae.

Type locality: SDSM V 2015-1: Lawrence County, South Dakota (for more detailed locality information contact SDSM). The type locality was originally identified as SDSM V 751 in Weishampel & Bjork (1989: p. 57). However, that number was already allocated to another location that does not match the township and range information recorded on the specimen card for SDSM 8656. The authors of the present study contacted the current landowners, who generously provided access to the location where SDSM 8656 was originally recovered (Fig. 2). Detailed geographic and geologic information was recorded during that and subsequent visits (Fig. 3), and a new locality number, SDSM V 2015-1, was designated for the type locality of D. lakotaensis.

Distribution: Lower Fuson Member, Lakota Formation, Inyan Kara Group, northern Black Hills region, South Dakota, USA.

Emended diagnosis of Dakotadon lakotaensis:Dakotadon lakotaensis displays two traits here identified as autapomorphies, though it should be noted that appropriate comparative material is not available for all closely related taxa and some of these characters may latter be found to be more widely distributed: (1) presence of a triangular projection along the dorsal surface of the lacrimal that inserts into the ventral margin of the prefrontal; and, (2) contact between the jugal and ectopterygoid consists of a medially projecting boss on the jugal that bears articulation surfaces for the ectopterygoid dorsally, medially, and posteroventrally.

This taxon is also differentiated from all non-hadrosauriform ankylopollexians by the following unique combination of characters: (1) absence of a diastema at the anterior end of the dentary tooth row (present in Barilium dawsoni); (2) dentary tooth row straight in lateral view (convex in Owenodon hoggii); (3) caudal-most extent of tooth row medial to coronoid process but still rostral to the longitudinal axis of the process (situated level with the longitudinal axis or further posterior in Lanzhousaurus magnidens and Fukuisaurus tetoriensis); (4) straight ventral margin of the anterior portion of the dentary leading to the predentary articulation (ventral margin inflected ventrally in Hippodraco scutodens); (5) dentary portion of coronoid process caudally inclined (vertical in Fukuisaurus tetoriensis and Barilium dawsoni); (6) dorsal expansion of coronoid process absent (present in Fukuisaurus tetoriensis and Barilium dawsoni); (7) two large denticles on each premaxilla (one on each premaxilla in Camptosaurus dispar and Theiophytalia kerri); (8) rostrodorsal process of maxilla present (absent in Camptosaurus dispar); (9) ventral margin of maxillary tooth row concave in lateral view (straight in Camptosaurus dispar, Cumnoria prestwichii, and Hippodraco scutodens); (10) ascending process of maxilla rostrocaudally broad and subtriangular in lateral view (rostrocaudally narrow and hook-like in Camptosaurus dispar and Cumnoria prestwichii); (11) antorbital fossa consists of a rostrocaudally elongate, elliptical depression restricted to the posterior half of the ascending process of the maxilla (occupies most of the lateral surface of the ascending process in Camptosaurus dispar and Cumnoria prestwichii); (12) anterior ramus of lacrimal tapers to a point (dorsoventrally expanded in Hippodraco scutodens and Theiophytalia kerri); (13) presence of a large neurovascular foramen on the medial surface at the base of the postorbital process of the jugal (absent in Camptosaurus dispar, Cumnoria prestwichii, and Fukuisaurus tetoriensis); (14) absence of a mediolaterally compressed, ‘blade-like’ anterior process of the squamosal (present in Iguanacolossus fortis); (15) posterior surface of the supraoccipital anterodorsally inclined (vertical in Lurdusaurus arenatus); (16) presence of a rostrocaudally directed groove along the ventral surface of the basioccipital (absent in Lurdusaurus arenatus); (17) surface between the basipterygoid processes of the basisphenoid smooth (presence of a sharply defined ridge in Lurdusaurus arenatus, Uteodon aphanoecetes, and Cumnoria prestwichii and a ventrally directed prong in Camptosaurus dispar); (18) marginal denticles on dentary teeth ‘tongue-shaped’ with smooth edges (‘tongue-shaped’ with mammillated edges in Barilium dawsoni, Lanzhousaurus magnidens, and Fukuisaurus tetoriensis); (19) dentary teeth bear parallel and similarly prominent primary and secondary ridges with multiple faint accessory ridges arising from marginal denticles (prominent primary ridge and multiple faint accessory ridges on either side in Lanzhousaurus magnidens); (20) maxillary teeth bear primary ridges and multiple parallel accessory ridges on either side (accessory ridges restricted to the mesial side of the primary ridge in Fukuisaurus tetoriensis).

Several non-hadrosauriform ankylopollexian taxa are known from such fragmentary material that comparisons to D. lakotaensis are limited or entirely impossible. For three taxa, Osmakasaurus depressus, Delapparentia turolensis, and Planicoxa venenica, the only material preserved in common with D. lakotaensis are the caudal vertebrae. In D. lakotaensis a pair of narrow ridges run anteroposteriorly from the anterior articulation facets for the chevrons to the posterior articulation facets. No such structures are present in O. depressus or D. turolensis, but similar ridges are present in H. fittoni and P. venenica and were previously reported as an autapomorphy of the latter taxon (DiCroce & Carpenter, 2001). Draconyx loureiroi also lacks the ventral ridges on the caudal vertebrae, but the preserved maxillary tooth crowns are indistinguishable from those of D. lakotaensis. Overlapping material is not preserved for D. lakotaensis and Cedorestes crichtoni, preventing direct comparison and differentiation of these two taxa that are presumed to be closely related (McDonald et al., 2010: Fig. 39).

Comments:Weishampel & Bjork (1989) provided a diagnosis for D. lakotaensis composed of eight characters: (1) supraoccipital incised beneath parietal and squamosals; (2) loss of median ridge on the supraoccipital; (3) single aperture for both branches of the facial nerve; (4) relatively large antorbital fenestra; (5) loss of contact between the maxilla and lacrimal at the jugal-maxilla articulation; (6) relatively small maxillary and dentary teeth; (7) few maxillary tooth families combined with low tooth density; and, (8) reduced z-spacing and longer wave of alternating teeth from the back of the jaws (W) in the mandibular dentition. While many of those features are unique in the context of D. lakotaensis being a hadrosauriform (i.e., referred to Iguanodon), many of those characters are now uninformative given the more recent recovery of this species as a non-hadrosauriform ankylopollexian. Several of those characters are also shown to be the result of postmortem deformation of SDSM 8656, eliminating their diagnostic utility. The perceived incision of the supraoccipital beneath the parietal and squamosals is an artifact of deformation in SDSM 8656 and not a true feature. Further preparation of the supraoccipital also reveals that a dorsoventrally oriented, transversely broad median ridge is present on the supraoccipital. The presence of a single aperture for the facial nerve, while different from the condition seen in some other taxa previously referred to Iguanodon (e.g., Mantellisaurus atherfieldensis), is plesiomorphic for Iguanodontia and not unexpected given the current systematic relationships of D. lakotaensis (McDonald, 2012; this study). The size of the antorbital fenestra in D. lakotaensis, while larger than some of its close relatives, is intermediate in size between the conditions seen in Camptosaurus and Iguanodon, as would be expected. Loss of contact between the maxilla and the ventral ramus of the lacrimal is also not unusual and occurs in other closely related taxa (i.e., Theiophytalia kerrii). The remaining three characters, all related to the dentition, describe conditions that are relatively common among most non-hadrosauriform ankylopollexians, most of which were unknown at that time. Therefore, none of these characters are autapomorphies of D. lakotaensis, although some of them are useful in part for differentiating this species from other closely related taxa (see Diagnosis above).

Paul (2008), in removing SDSM 8656 from Iguanodon and into the new taxon Dakotadon, provided an emended diagnosis for the newly combined Dakotadon lakotaensis: (1) ventral margin of premaxilla not below maxilla, maxillary process of premaxilla deep; (2) dorsal midline trough in nasals; (3) dorsal apex of maxilla near middle of element; (4) antorbital fossa and fenestra large; (5) lacrimal long, does not contact maxilla posterior to antorbital fossa; (6) dentary moderately deep, diastema absent; and, (7) nineteen tooth positions in maxilla. Characters 1, 2, 3, 4, and 7 were considered unambiguous autapomorphies of D. lakotaensis in that study. Character one is inaccurate, as the ventral margin of the premaxilla does extend slightly below the maxilla in SDSM 8656. The dorsal midline trough in the nasals is at least accentuated by postmortem deformation, and even if a slight trough was present, a similar feature also occurs in Theiophytalia kerri (Brill & Carpenter, 2007). Character three is confusing as presented because the posterior portion of the maxilla was incompletely known prior to this study, so the exact position of the dorsal apex could not have been determined. Additionally, the anterior and middle portions of the preserved left maxilla of Theiophytalia kerri is very similar to D. lakotaensis, indicating that the overall shape of this bone in the latter taxon is not unique. As noted above, the size of the antorbital fossa and fenestra in D. lakotaensis is not unexpected given its systematic position and the lack of contact between the lacrimal and maxilla posterior to the antorbital fossa is not restricted to D. lakotaensis. The presence or absence of a dentary diastema varies between the left and right side in SDSM 8656 (see description below), excluding its use as a diagnostic feature of D. lakotaensis. Damage to the posterior portion of the maxilla makes it uncertain if there were 19 or 20 tooth positions (see description below), but this interpretation was only possible once the posterior portion of the left maxilla was discovered in this study. Also, there are few other taxa that preserve a complete maxilla for comparison purposes. Thus, none of the proposed autapomorphies of Paul (2008) diagnose D. lakotaensis and the full set of characters insufficiently differentiates this taxon from other ankylopollexians.

Description of the Skull of Dakotadon Lakotaensis

The only detailed description of Dakotadon lakotaensis was provided by Weishampel & Bjork (1989). In 2014, the holotype of D. lakotaensis, which was never fully cleaned prior to description, underwent extensive preparation that exposed new regions of the skull and clarified the overall morphology. Given that extensive preparation work and the increased diversity of basal ankylopollexian taxa described since the original description of D. lakotaensis, a full redescription and comparison of the holotype is provided herein.

Cranium

Premaxilla

Portions of both premaxillae are preserved (Figs. 4 and 5B). The left premaxilla is missing the dorsal process, part of the border of the subnarial fossa, and a small section of the posterolateral corner of the oral margin (Fig. 4A). The right premaxilla is less complete, missing the dorsal process, the posterior half of the lateral oral margin, and the posterior-most portion of the posterolateral process (Fig. 4B). The right premaxilla is more transversely crushed than the left, so much of this description is based on the left premaxilla. The premaxillae remain unfused, although they are tightly appressed, especially anteriorly. The premaxillae are edentulous and the oral margin is offset ventrally slightly below the oral margin of the maxilla (Fig. 4A). In ventral view the anterior margin of the premaxillae is bluntly rounded with a sharp angle between the anterior-most margin and the lateral margins (Figs. 4C and 5B). The anterodorsal surface is highly rugose and roughened where the rhamphotheca covered the premaxilla. In ventral view, four anteroposteriorly elongate denticles extend from the anterior margin, two on each premaxilla (Fig. 5B: pd). These four denticles would have tightly interlocked with three corresponding denticles on the predentary, forming a complex shearing surface. This differs from the condition seen in Theiophytalia kerri and Camptosaurus dispar where each premaxilla displays a single denticle (McDonald, 2011). Posterior to these denticles the ventral surface of the oral margin is first convex, then transitions to concave in the posterolateral corner (Fig. 5B). The posterolateral corner of the oral margin is angular and projects further laterally than the anterior end of the maxilla. A raised ridge is present on the ventral surface along the midline, but this feature may be accentuated by the transverse crushing of the specimen. The posterior portion of the ventral surface is broadly concave and the posterior margin contacts the vomer, although the exact nature of this contact is unclear. The ventral surface of each premaxilla is pierced by two foramina (Fig. 5B). The first foramen is immediately posterior to and situated between the two prominent denticles. This foramen appears to connect with the pits and grooves on the anterodorsal surface of the premaxilla (Fig. 5B: pavf). The second foramen is situated directly posterior to the first foramen and is positioned at the anterior margin of the broad concavity in the ventral surface of the premaxillae (Fig. 5B: ppvf). The full path of this latter foramen cannot be traced.

The subnarial fossa is deeply inset in the lateral surface of the body of the premaxilla dorsal to the rim of the oral margin (Fig. 4A). This fossa is posterodorsally inclined, and the posterior end extends dorsal to the anterior-most end of the maxilla. The dorsal portion of this fossa is not preserved, and no foramina are observed in the preserved portion of the fossa. The posterolateral process arises from the posterolateral corner of the oral margin. This process forms the ventral and part of the posterior margin of the external naris. The posterolateral process is relatively broad dorsoventrally and extends posterodorsally along the lateral surface of the skull (Fig. 4A). The ventral margin forms an elongate contact with the anterodorsal surface of the maxilla. Unlike in some basal ornithopods (e.g., Thescelosaurus neglectus: Boyd, 2014) the maxilla does not insert into the posterior margin of the premaxilla. The dorsal margin of the posterolateral process posterior to the external naris forms an elongate contact with the nasal. The base of the posterolateral process is transversely broad where it contacts the maxilla, but this process gradually narrows as it extends posteriorly, until it is simply a thin sheet overlapping the lateral surface of the anterior-most portion of the lacrimal. The posterior-most tip tapers to a broadly rounded point that contacts the nasal, prefrontal, and lacrimal (Fig. 5A).

Nasal

The majority of both nasals are preserved, but they are highly crushed and distorted. The anterior margin of each nasal is concave, creating an anteriorly projecting point over the nares along the midline. The anterior margin is also slightly rugose. The ventrolateral margin forms an elongate contact with the posterolateral process of the premaxilla. The posteromedial surface of the nasal contacts the lacrimal, but the premaxilla and prefrontal exclude the nasal from contacting the lacrimal on the exposed lateral surface of the skull. The posterior margins of both nasals are damaged, obscuring the contact with the frontals. The dorsal surface of the nasal is pierced by a few small foramina, the position and number of which varies on each side.

Paul (2008) described the presence of an elongate midline trough in the nasals as an autapomorphy of Dakotadon lakotaensis. However, the lateral portions of the nasals have been rotated so they are dorsally inclined, while the medial portions are fractured apart from the lateral portion (left side) or distorted (right side) and crushed into the nasal cavity. As a result, the prominent trough currently present in this specimen is at least part, if not wholly, the result of postmortem distortion. A similar trough is present in the holotype of Theiophytalia kerri, which is also transversely crushed (Brill & Carpenter, 2007). Brill & Carpenter (2007) suggest that if the nasals were dorsally arched with a slight to moderately developed midline trough, the observed postmortem distortion may have simply accentuated the trough in these specimens. Therefore, a midline trough in the nasals in these taxa is either entirely a result of deformation, or is a more broadly distributed character than noted by Paul (2008), excluding the use of this feature as an autapomorphy of D. lakotaensis.

Lacrimal

Most of the left lacrimal is preserved in original position and the anterior-most portion of the right lacrimal is present and slightly displaced. The lacrimal forms part of the lateral surface of the nasal cavity, the anterior margin of the orbit, and the dorsal and posterior margins of the antorbital fenestra (Fig. 5A). The anterior portion of the lacrimal is broadly overlapped by the premaxilla, obscuring much of the morphology. The dorsolateral surface of the lacrimal forms the ventral portion of the articulation surface for the supraorbital, with the ventral margin of this contact demarcated by an anteroposteriorly elongate ridge. Ventral to this articulation surface and dorsal to the antorbital fenestra the lateral surface of the lacrimal displays a broad depression. Just anterior to the antorbital fenestra the ventral margin of the lacrimal is grooved to form the dorsal portion of the antorbital fossa. Anterior to the lacrimal portion of the antorbital fossa the ventrolateral surface of the lacrimal is overlapped by a dorsal extension of the maxilla. The ventral process of the lacrimal is angled posteroventrally to the contact with the jugal. The anteromedial surface of that process is concave along the margin of the antorbital fenestra (Fig. 6). The ventromedial corner of the ventral process contacts the anterolateral tip of the palatine, and this contact surface is shallowly grooved, indicating the presence of a small fenestra along this contact. A large excavation in present in the posterior surface of the lacrimal, forming the lacrimal canal. The posteromedial margin of the lacrimal forms a short wing that extends medially, forming the anteromedial border of the orbit (Fig. 6).

Maxilla

The maxilla is roughly triangular in lateral view, with a ventrally concave oral margin and ventrally extending anteroventral and posteroventral corners (Fig. 4A). There are short edentulous regions both anterior and posterior to the maxillary tooth row (Fig. 4A). In dorsal view, the lateral surface of the maxilla is slightly concave, and in ventral view the tooth row is slightly bowed medially, with the anterior and posterior ends curving laterally (Fig. 4C). The rostroventral process at the anterior end of the maxilla curves ventrally as in Theiophytalia kerri, but unlike the rostrally projected process present in Camptosaurus dispar (Brill & Carpenter, 2007; McDonald, 2011). There is no pronounced ridge or shelf on the lateral surface dorsal to the maxillary tooth row, unlike in Cumnoria prestwichii (Galton & Powell, 1980), although there are a series of foramina along the lateral surface that form a continuous row just dorsal to the tooth row. The posterior-most of these foramina is the largest (Fig. 5A: pmf) and it connects medially with a fenestra formed between the maxilla, jugal, ectopterygoid, and palatine (Fig. 6: pmf). The anterodorsal surface participates in a long contact with the posterolateral process of the premaxilla. The dorsal process of the maxilla is somewhat anteroposteriorly expanded, unlike the narrower, posteriorly-curved process seen in Cumnoria prestwichii and Camptosaurus dispar (McDonald, 2011). The dorsal process overlaps a small portion of the posterolateral process of the premaxilla anteriorly and the anterior portion of the lacrimal posteriorly (Fig. 5A). The caudal half of the dorsal process of the maxilla is indented by the ‘D-shaped’ antorbital fenestra, and the lateral surface of the maxilla anterior to this indentation bears much of the shallow, ovoid antorbital fossa (Fig. 5A). Posterior to the antorbital fenestra the maxilla does not contact the lacrimal, instead forming a long, sinuous scarf joint with the anterior process of the jugal (Figs. 4A and 5A). The posterior end of the maxilla is bifurcated dorsal to the last two maxillary tooth positions. The posterodorsal process of the maxilla is relatively short, while the posteroventral process is more elongate and curves ventrally, extending below the level of the occlusal surface of the maxillary tooth row. The ectopterygoid inserts into the resulting groove between these two processes and makes extensive contact with the dorsolateral surface of the posteroventral process of the maxilla. Much of the medial surface of the maxilla is obscured by matrix and the vomer. On the anteromedial surface a slight, medially projecting shelf is present dorsal to the tooth row, but its extent is obscured by crushing. The morphology and extent of any contact between the maxillae and the vomer is unknown. The posteromedial surface of the maxilla bears an extensive, anterodorsally inclined articulation facet for the palatine (Fig. 6). Anterior to that articulation facet and posteromedial to the antorbital fenestra is a dorsally concave shelf is present that is pierced by an anteroposteriorly elongate foramen that extends into the body of the maxilla (Fig. 6). Just dorsal to the maxillary tooth row a series of foramina pierce the medial surface of the maxilla.

Jugal

The anterior process of the left jugal is preserved in articulation with the maxilla, lacrimal, ectopterygoid, and palatine. The anterior-most tip of this process extends forward to form a small portion of the posteroventral corner of the antorbital fenestra (Fig. 5A), although the portion that reaches the antorbital fenestra is dorsoventrally narrower than in Theiophytalia kerri (Brill & Carpenter, 2007). The dorsal surface bears a relatively abbreviated, dorsally oriented articulation facet for the lacrimal, while the ventral margin forms an elongate, anteroventrally facing facet for the maxilla (Fig. 5A). The latter facet is anterodorsally inclined along its length, and is slightly sinuous. Medially, there is an anteroposteriorly elongate ridge near the anterior end, and the articulation facet for the palatine is just ventral to this ridge. Medial to the posteroventral corner of the orbit a medially projecting boss is present that bears the articulation facet for the ectopterygoid dorsally, medially, and posteroventrally. This boss does not contact the maxilla. The contact between the jugal and ectopterygoid present in this specimen differs from the condition seen in Fukuisaurus tetoriensis where the ectopterygoid contacts the posterior surface of a roughly dorsoventrally oriented ridge (Kobayashi & Azuma, 2003: Fig. 3). Alternatively, in Cumnoria prestwichii the ectopterygoid articulation facet is dorsally situated on a medial projection of the jugal, although the articulation does not wrap around the medial and posteroventral surfaces (Galton & Powell, 1980: Fig. 1G). A sharp ridge extends posteriorly from the medial jugal boss to the broken margin of the jugal. A small foramen is present on the medial surface of the jugal just ventral to that ridge that pierces straight through to the lateral surface of the jugal. A similar foramen is seen on the medial surface of the jugal at the base of the postorbital process in more derived ankylopollexians (e.g., Mantellisaurus atherfieldensis and Ouranosaurus nigeriensis; McDonald, 2011). Weishampel & Bjork (1989) also reported the presence of a separate piece they identified as the dorsal portion of the postorbital process of the jugal and the ventral process of the postorbital. The location of this piece is currently unknown. A cast believed to represent this piece was stored with other casts of this specimen, but it is ambiguous as to the identity of the elements represented. The remaining portions of the jugal are missing.

Prefrontal

Much of the left prefrontal is preserved, but is crushed, faulted, and somewhat distorted. Much of the material previously identified as the posterior process of the prefrontal (e.g., Weishampel & Bjork, 1989: Fig. 1) actually belongs to the supraorbital, a fact that was made clear after recent preparation efforts. Additionally, interpretation of the morphology of the anterior portion of the prefrontal and its contacts with the nasal, lacrimal, and premaxilla has varied in prior publications. Weishampel & Bjork (1989) reconstructed the prefrontal contacting the nasal, posterolateral process of the premaxilla, and the lacrimal. They also reconstructed the contact with the lacrimal as along a roughly anteroposteriorly oriented straight line, with the posterodorsal corner of the lacrimal missing. Brill & Carpenter (2007) reconstructed the lacrimal excluding the prefrontal from contacting the posterolateral process of the premaxilla and the contact between the prefrontal and lacrimal along an anterodorsally inclined line. The reconstruction in Paul (2008) is similar to that of Brill & Carpenter (2007) except that the inclined suture between the lacrimal and prefrontal is more sharply inclined anteriorly, making a ventrally convex contact. Portions of all of these interpretations differ from that presented herein as detailed below.

The prefrontal contacts the lacrimal ventrally, the nasal anterodorsally, the posterolateral process of the premaxilla at its anterior tip, the frontal medially and posteriorly, and bears much of the articulation surface for the supraorbital on its ventrolateral surface. The anterior portion of the prefrontal is preserved in situ, although the dorsal-most portion is missing (Fig. 5A). The contact between the lacrimal and the prefrontal is largely along an anteroposteriorly oriented line, but slightly posterior to the middle of this suture there is a triangular projection of the lacrimal that inserts dorsally into the prefrontal (Fig. 5A: white dashed line). There is no evidence that the presence of this projection is taphonomic, but this projection is limited to the lateral margin and does not continue to the medial margin of the lacrimal/prefrontal contact. This complex contact is not known from any other basal iguanodontian, although many described species do not preserve this region of the skull. Prior confusion regarding the morphology of this contact likely resulted from the combined presence of prominent fractures running through both the lacrimal and prefrontal, some of which match the position of the previously reconstructed contacts, the unusual morphology of the contact, and the fact that matrix was not fully removed from this region previously.

The anterior end of the prefrontal is overlapped laterally by the posterolateral process of the premaxilla and the nasal. The lateral surface of the anterior portion is relatively flat. The anterodorsal portion is missing. Along the orbital margin the prefrontal contacts the lacrimal just dorsal to the lacrimal canal. Medially, the anterior portion of the prefrontal is concave both dorsoventrally and anteroposteriorly (Fig. 6). The prefrontal forms the anterodorsal corner of the orbit, and the orbital margin was relatively smooth.

The middle portion of the prefrontal is present on the section of the specimen that preserves the anterior portion of the skull roof (Figs. 7A–7C). Here, the middle portion of the prefrontal has been faulted underneath the posterior portion of the prefrontal and the anterior-most portion of the frontal, medial to the preserved position of the supraorbital (Fig. 7B). Similarly, the posterior portion of the prefrontal has been slightly thrust back into the dorsal surface of the frontal, resulting in an area of slightly crushed and displaced bone that obscured the contacts between these elements prior to more thorough preparation of the specimen (Figs. 7A and 7C). In addition to obscuring the contacts between the prefrontal and the frontal, this deformation also artificially shortened reconstructions of both the overall skull length and the anteroposterior length of the orbit. Instead of having a dorsoventrally tall and anteroposteriorly narrow orbit (as is seen in Iguanodon bernissartensis; Paul, 2008), the orbit was more anteroposteriorly elongate, as reconstructed for Camptosaurus dispar (Brill & Carpenter, 2007: Fig. 3.3) and Hippodraco scutodens (McDonald et al., 2010: Fig. 21). The medial contact between the frontal and the prefrontal is not preserved; however, the contact between the posterior process of the prefrontal and the frontal consists of a tongue and groove contact, with the frontal inserting into the prefrontal and the prefrontal overlapping the frontal dorsally and ventrally.

Postorbital

Much of the dorsal portion of the left postorbital is preserved (Figs. 7A–7C), while only a small piece of the right postorbital is preserved along the contact with the parietal and frontal. A previously described piece containing the jugal process of the left postorbital cannot be located at this time and a cast of that piece housed with the specimen is difficult to interpret, so that portion is excluded from this description. This description does include a new piece of the specimen recently prepared and identified that contains the posterior process of the postorbital and the anterior process of the squamosal and fits onto the previously described portion (Fig. 7C).

The postorbital formed the posterodorsal corner of the orbit, the anterodorsal corner of the infratemporal fenestra, and the anterolateral corner of the supratemporal fenestra. The body of the postorbital is laterally concave and the majority of the lateral surface is slightly roughened. A slight rugose boss projects laterally and anteriorly into the orbit that may have formed a contact for either the supraorbital or for connective tissues attached to the supraorbital (Fig. 7C). The anterior process is anteroposteriorly short and mediolaterally broad, extending ventral to the frontal. Thus, the articulation surface for the frontal on the postorbital is dorsomedially facing, unlike the medially facing facet seen in some taxa (e.g., Thescelosaurus neglectus; Boyd, 2014). The articulation surface between the frontal and postorbital consists of a series of interlocking ridges and grooves that are roughly mediolaterally oriented. The postorbital wraps around the entire posterolateral corner of the frontal, with a medial projection extending to contact the anterolateral corner of the parietal (Figs. 7A and 7B). Just ventral to the contact with the parietal on the medial surface a concave socket is present that supports part of the dorsolateral head of the laterosphenoid (Fig. 7B). The posterior process is incompletely preserved, but enough is present to show that the ventromedial margin of the process possessed a deep groove for receipt of the anterior process of the squamosal (Figs. 7A and 7B). The anteroventral surface of the postorbital is concave, forming part of the medial wall of the orbit.

Frontal

The left frontal is incomplete anteriorly and the prefrontal is crushed into the anterolateral margin (Fig. 7C). Additionally, a fracture runs through the left frontal from the posterior margin near the contact with the postorbital and parietal anteromedially to the midline suture with the right frontal. The portion of the skull roof anterolateral to this fracture is slightly pushed posterodorsally and rotated clockwise in dorsal view (Fig. 7A). As a result, the orbital margin of the left frontal artificially appears to angle more strongly anteromedially than it was naturally. The preserved portion of the right frontal is relatively undeformed, but the lateral and anterior portions are missing.

The frontals contact the parietals posteriorly, the postorbital posterolaterally, the laterosphenoid posteriorly ventral to the contact with the parietal, and the prefrontal anterolaterally just anterior to the orbital margin (Figs. 7A and 7B). The contact with the nasals is not preserved. The frontals contact each other along an interdigitating suture along the midline of the skull roof. The frontals are broadly concave dorsally, with a slight ridge present along the midline suture. The orbital margin of the frontals is relatively dorsoventrally thin and striated (Fig. 7C). The exact length of the orbital margin is uncertain because crushing on the left side has resulted in some faulting and overlap of portions of the prefrontal and frontal. The frontal contacts the prefrontal along a tongue and groove surface, where a thin plate of the frontal inserts into the posterior end of the prefrontal, with the prefrontal overlapping the frontal dorsally and ventrally. It is uncertain if this same contact was present along the medial surface of the prefrontal. The contact with the postorbital spans the entire posterolateral corner of the frontal and consists of a series of interlocking ridges and grooves. The postorbital also extends ventral to the frontal, contacting the laterosphenoid (Fig. 7B). The ventrolateral surface of the frontal is concave, forming the dorsal surface of the orbit.

The articulation surface for the laterosphenoid spans the posteromedial corner of the ventral surface, extending laterally to connect with the articulation surface for the postorbital. At the anteromedial margin of the contact surface for the laterosphenoid a sharp ridge arises on the ventral surface of the frontal. This ridge extends anterolaterally and borders a deep concavity along the midline of the frontals which housed the paired olfactory tracts (Fig. 7B: ot). The full extent and morphology of these tracts is not preserved. The posterior-most margin of the frontals, dorsal to the articulation for the laterosphenoid, forms an extensive transverse suture with the anterior margin of the parietal (Fig. 7A). The posterior margin of each frontal is slightly convex posteriorly, although not to the extent seen in Cumnoria prestwichii or in the basal ornithopods Hypsilophodon foxii, Thescelosaurus assiniboiensis, and T. neglectus (Galton, 1974; Galton & Powell, 1980; Brown, Boyd & Russell, 2011; Boyd, 2014).

Parietal

The parietals are indistinguishably fused, creating a single ‘saddle-shaped’ element. The majority of the parietal is preserved, although the right posterolateral corner and a middle section of the sagittal crest are missing. The parietal forms the medial margins of the supratemporal fenestrae, as well as part of the anterior and posterior margins. The anterior margin of the parietal contacts the slightly transversely convex posterior margin of the frontal along an extensive suture (Fig. 7A). The anterolateral corners of the parietal contact the medial processes of the postorbitals along a laterally concave articulation surface that results from the anteroventral corner extending further laterally than the anterodorsal corner. The anteroventral margins make extensive contact with the dorsal margins of the laterosphenoids. Near midlength along the ventral margin a ventromedially projecting ridge is present. The anterior end of this ridge marks the beginning of the contact with the anterodorsal margin of the supraoccipital, dorsal to the prootic. The posterolateral margins form a sinuous contact with the medial processes of the squamosals (Fig. 8B). The posteroventral corners of the parietal project posterolaterally, extending dorsal to the fused opisthotic/exoccipitals (Fig. 8B).

Figure 8 Braincase of Dakotadon lakotaensis (SDSM 8656).

(A) photograph of skull roof and braincase in left lateral view taken at an angle looking slightly posteriorly; (B) photograph of the braincase in posterior view; (C) photograph of the posterior portion of the braincase in left lateral view. Abbreviations: bo, basioccipital; bs, basisphenoid; bt, basipterygoid process; ci, crista interfenestralis; CN, cranial nerve; cpr, crista prootica; ct, crista tuberalis; dor, dorsal; fo, fenestra ovalis; fm, foramen metoticum; fr, frontal; gCN V1 & vcms, groove for the ramus ophthalamicus and the vena cerebralis media secunda; gCN V IIp, groove for the palatine ramus of CN VII; lat, lateral; ls, laterosphenoid; op, fused opisthotic/exoccipital complex; os, orbitosphenoid; par, parietal; pf, prefrontal; po, postorbital; pop, paroccipital process; pp, preotic pendant; post, posterior; pro, prootic; pt, pterygoid; ptf, post-temporal foramen; so, supraoccipital; sor, supraorbital; sq, squamosal; vc, Vidian canal; vcm, foramen for the vena cerebralis media; vent, ventral; vp, vena parietalis. Scale bars equal 5.0 cm.

The ventral surface of the parietal is deeply excavated for receipt of the supraoccipital. In posterior view there is a dorsoventrally tall and relatively transversely narrow gap between the lateral walls of the parietal into which the supraoccipital is situated (Fig. 8B). The posteroventral surface of the parietal capped the supraoccipital and formed an extensive contact with that element. A small ventral projection is present along the midline at the posterior margin that indented the dorsal margin of the supraoccipital (Fig. 8B). In dorsal view the posterior margin is deeply concave, although this is in part owing to the transverse crushing in this specimen. The lateral margins are also broadly dorsolaterally concave where they form the medial walls of the supratemporal fenestrae, giving the parietal an ‘hour-glass’ shape. Slight ridges arise along the frontoparietal contact and extend in a laterally concave arc posteromedially towards the midline of the parietal. Damage to the specimen makes it uncertain if these ridges merged to form a single sagittal crest as is seen in some taxa (e.g., Hippodraco scutodens; McDonald et al., 2010) or if they remained slightly separated along their length as occurs in some specimens of Thescelosaurus neglectus (e.g., specimen TLAM.BA.2014.027.0001). Near the posterior end of the element these ridges diverge again (or separate if they do indeed form a single sagittal crest), extending towards the posterolateral corners as broad, rounded crests. Between these crests a small posteromedially situated sulcus is present on the dorsal surface.

Squamosal

Two separate pieces of the left squamosal are preserved. The anterior-most portion of the squamosal is preserved in articulation with the posterior process of the postorbital. This process is triangular in cross section and fits into a groove in the ventromedial surface of the posterior process of the postorbital. This process widens both dorsoventrally and mediolaterally as it extends posteriorly. The morphology of this contact is the same as described for Cumnoria prestwichii (Galton & Powell, 1980), but differs from the mediolaterally compressed, ‘blade-like’ process present in Dryosaurus altus, Iguanacolossus fortis, Tenontosaurus tilleti, and Zalmoxes robustus (McDonald et al., 2010).

The medial process of the left squamosal is preserved in articulation with the left parietal, the fused opisthotic/exoccipital complex, supraoccipital, and prootic (Figs. 8A–8C). The squamosals were broadly separated from each other by the parietal. The medial process projected anteromedially, although this has been accentuated by lateral crushing of the specimen. Enough of the medial process is preserved to describe its contacts with the braincase elements. The medial margin makes extensive contact with the parietal, with the squamosal overlapping the lateral surface of the posterolateral portion of the parietal. In lateral view, this contact is sinuous, with a short process along the margin of the parietal projecting into a groove in the squamosal (Fig. 8C). In posteromedial view the contact between the parietal and squamosal forms an anteromedially convex curve. This curve results from the extension of a ‘finger-like’ process projecting from the posteroventral corner of the parietal that extends further posteriorly than the rest of the parietal (Fig. 8B). Ventral to the articulation surface for the posteroventral process of the parietal, a low ridge is present on the ventromedial margin of the medial process of the squamosal that separates that contact from the articulation surface for the fused opisthotic/exoccipital on the ventral surface. The latter articulation surface is concave ventrally to fit tightly against the fused opisthotic/exoccipital. The medial margin of the anteroventral corner of the medial process of the squamosal possesses a small articulation surface that may have fit against the posterolateral corner of the supraoccipital, but the bones are not currently in contact. The lateral margin of the anteroventral corner of the medial process of the squamosal forms a short contact with the posterodorsal margin of the prootic (Fig. 8C).

Palatoquadrate

Pterygoid

A small portion of the quadrate alar process of the left pterygoid is preserved in articulation with the left basipterygoid process of the basisphenoid (Figs. 8A and 8C: pt). The preserved portion is broadly curved to accommodate the basipterygoid process, being concave dorsomedially.

Palatine

The body (ventrolateral) portion of the left palatine is preserved in contact with the maxilla, jugal, and ectopterygoid (Fig. 6: pal). The anteroventral surface of the palatine is broadly cupped where it makes extensive contact with posteromedial surface of the maxilla. The anterior-most tip of the palatine extends dorsolaterally where it contacts the ventromedial corner of the lacrimal. The anterior tip of the palatine is slightly damaged, but the ventral surface of the lacrimal is excavated by a narrow groove, indicating that a small fenestra was present at the contact between the lacrimal and the palatine. The anterolateral margin of the palatine contacts an anteroposteriorly elongate, dorsoventrally narrow facet on the medial surface of the jugal that is situated anterior to the medial jugal boss. This creates a fenestra between the jugal, palatine, ectopterygoid, and maxilla (Fig. 6: pmf). That fenestra extends ventrolaterally through the maxilla where it emerges as a large foramen on the lateral surface of the maxilla ventral to the contact between the maxilla and jugal (Fig. 5A: pmf). Posterior to the small postpalatine foramen (Fig. 6: ppf), the posterolateral surface of the palatine contacts the anteromedial margin of the ectopterygoid. The remainder of the palatine is missing.

Ectopterygoid

The left ectopterygoid is incompletely preserved in articulation with the jugal, maxilla, and palatine (Fig. 6: ep). The posterior-most portion is absent, and sections of the lateral process are missing (Fig. 6: dam). The lateral-most end of the ectopterygoid contacts a medially-projecting boss on the maxillary process of the jugal that is positioned just dorsal to the posterior-most contact between the jugal and the maxilla. A ‘tab-shaped’ process of the ectopterygoid extends dorsolaterally from the lateral end of the ectopterygoid, overlapping the dorsal surface of the medial jugal boss. Ventral to this process, the ectopteryoid makes extensive contact with the medial and ventral surfaces of the jugal, and this contact extends ventrally to the contact between the jugal and the maxilla. The entirety of the ventromedial surface of the preserved portion of the ectopterygoid contacts the posteroventral-most projection of the maxilla dorsal to the posterior-most alveolus in the maxilla, which results in a continuous contact between the ectopterygoid, maxilla, and jugal in this area. A small postpalatine fenestra is present between the anteromedial margin of the ectopterygoid, the posterolateral margin of the palatine, and the posterodorsal surface of the maxilla (Fig. 6: ppf). Posterior to the postpalatine fenestra the anteromedial margin of the ectopterygoid contacts the posterolateral margin of the palatine, although the extent of this contact is obscured by damage and crushing. The nature of the contact between the ectopterygoid and the pterygoid is unknown in this specimen.

Vomer

The majority of the vomer is preserved in original position, although it has been damaged and distorted by the transverse crushing of the specimen. The anterior end is transversely expanded where it makes contact with the posterior margins of the premaxillae (Fig. 5B: vo). The lateral margins of the anterior end likely contact the anteromedial surfaces of the maxillae, but this cannot be confirmed. The vomer becomes transversely narrower toward the posterior end while expanding dorsoventrally (Fig. 6: vo). A deeply incised groove is present on the dorsal surface beginning at the posterior end (anterior extent of this groove not exposed), giving the posterior portion of the vomer a ‘y-shaped’ transverse cross section. A small piece of bone is preserved within this groove at the posterior-most end of the vomer. This may be a piece of the anterior-most portion of the pterygoid, as is seen in other neornithischians like Thescelosaurus neglectus (Boyd, 2014), but given that most of the pterygoids are not preserved in this specimen it is impossible to be certain. The dorsomedial portions of the palatines are not preserved, making it uncertain if the palatines contact the posterolateral surfaces of the vomer.

Braincase

This specimen includes one of the most well-preserved and complete braincases of any non-hadrosauriform ankylopollexian taxon (Fig. 8). Thus, detailed description of this region is crucial to understanding the evolution of the braincase within Ornithopoda, especially the transition between basal ornithopods (e.g., Hypsilophodon foxii) and derived hadrosauroids. Additionally, recent preparation of this specimen provides clarity with regard to the position of various contacts and cranial nerve (hereafter abbreviated CN) foramina that were previously uncertain or incorrectly identified.

Basioccipital

The basioccipital contacts the fused opisthotic/exoccipitals dorsolaterally and the basisphenoid anteriorly. Posteriorly, the basioccipital forms the majority of the occipital condyle, with small contributions from the fused opisthotic/exoccipitals. The posterodorsal surface is indented to form a small portion of the foramen magnum, and a broad groove extends anteriorly along the dorsal surface to form the floor of the braincase. The basioccipital portion of the floor of the braincase is not fully visible owing to crushing and remaining matrix. The occipital condyle angles posteroventrally and the articular surface extensively wraps around the lateral and ventral margins of the basioccipital, with a pronounced lip present anteriorly along the lateral and ventral margins (Fig. 8C). The occipital condyle is also relatively short anteroposteriorly compared to its dorsoventral height. Overall, the occipital condyle most closely resembles that of Uteodon aphanoecetes than of any other taxon (McDonald, 2011: Fig. 7B). In fact, the occipital condyle of this specimen extends further ventrally than the basal tubera, which was reported as the lone autapomorphy of Uteodon aphanoecetes by McDonald (2011), but in SDSM 8656 the presence of this feature is likely the result of postmortem crushing and displacement of some portions of the braincase. The bone surface immediately anteroventral and anterior to the left margin of the occipital condyle is damaged. A deep, anteroposteriorly oriented groove is present on the anterior half of the ventral margin of the basioccipital. Crushing and slight distortion of the area between the basal tubera makes it impossible to tell if an anteroposteriorly oriented sharp ridge was present. The anteroventral corners of the basioccipital flare ventrally and laterally to form the posterior portions of the bases of the basal tubera (Fig. 8C: bt).

Basisphenoid/Parasphenoid

In most ornithischians the basisphenoid and the parasphenoid are indistinguishably fused (Galton, 1989), making it difficult to determine where the two elements meet. The anterior portion of what would be the fused basisphenoid/parasphenoid is missing, exposing the sella turcica and obscuring the morphology of the cultriform process. Thus, the parasphenoid contribution to this element is considered lost and is not discussed. The right lateral side of the basisphenoid is too damaged to provide much information, but the left lateral side and the ventral margin are well-preserved enough to provide information about the morphology of this element, although some crushing and distortion is present.

The posterior surface of the basisphenoid forms an extensive contact with the basioccipital, with the midline of the basioccipital inserting anteriorly into the basisphenoid and the posterolateral ends of the basisphenoid overlapping the lateral margins of the basioccipital, a condition also seen in the basal ornithopods Changchunsaurus, Haya, and Thescelosaurus neglectus (Jin et al., 2010; Makovicky et al., 2011; Boyd, 2014) and the basal iguanodontian Anabisetia (Coria & Calvo, 2002). The posterolateral corners of the basisphenoid are expanded laterally and ventrally, forming the basal tubera along with a small contribution from the basioccipital (Fig. 8C: bt). The preotic pendants are situated anterodorsal to the basal tubera on the lateral surface of the basisphenoid and are separated from the basal tubera by a narrow but deep groove. The close proximity of the preotic pendants and the basal tubera may be in part owing to the transverse crushing present in this specimen. The basipterygoid processes, of which only the left is preserved, arise from the ventrolateral margins of the basisphenoid anterior to the basal tubera. These processes extend ventrolaterally and slightly posteriorly from the basisphenoid and are situated much closer to the basal tubera than is seen in more basal ornithopods (Galton, 1989) or in other basal ankylopollexians (McDonald, 2011: Fig. 7). While this is in part owing to crushing in this specimen, the preserved base of the right basipterygoid process arises from the ventral surface of the basisphenoid closer anteroposteriorly to the basal tubera than in Uteodon aphanoecetes (McDonald, 2011: Fig. 7B), more closely resembling the condition seen in Cumnoria prestwichii (McDonald, 2011: Fig. 7C) and in basal hadrosauriforms (e.g., Mantellisaurus atherfieldensis: Norman, Hilpert & Hölder, 1987). Medial to the posterior margin of the basipterygoid processes, a narrow groove runs anterodorsally on the lateral surface of the basioccipital, connecting to the other lateral groove at the posteroventral margin of the preotic pendant. Where these two grooves meet there is a pronounced foramen that penetrates anteromedially into the basisphenoid (Figs. 8A and 8C: vc). That foramen is the exit of the Vidian canal through which passed the internal carotid artery and the palatine ramus (CN VIIp) of CN VII (facialis nerve), and this foramen extends into the ventrolateral corner of the sella turcica. The groove that extends dorsolaterally from the Vidian canal posterior to the preotic pendant leads to the foramen for CN VII (Fig. 8C: gCN V IIp). The ventral surface of the basisphenoid is slightly concave and sharp ridges arise from the anterior margins of the basipterygoid processes that continue anteriorly along the ventrolateral margins of the preserved portion of the basisphenoid.

In anterior view the basisphenoid is broken open to expose the inside of the sella turcica (not figured). The foramina for the Vidian canals penetrate the ventrolateral corners of the posterior surface of the sella turcica. A thin plate of bone forms the roof of the sella turcica and separates that region from the floor of the braincase. The basisphenoid portion of the floor of the braincase has two shallow, anteroposteriorly oriented grooves that each connect to two foramina that penetrate anteroventrally into the dorsal surface of the sella turcica. These foramina likely contained CN VI, as is the case in the basal ornithopods Thescelosaurus assiniboiensis and T. neglectus (Boyd, 2014). A groove extends anteriorly from each of these foramina along the lateral walls of the sella turcica that is bounded ventrally by a sharp ridge.

The dorsal margin of the basisphenoid contacts the fused opisthotic/exoccipitals, prootic, and laterosphenoid. The anterodorsal surface of the basisphenoid contacts the posteroventral margin of the orbitosphenoid (Fig. 8A). At the dorsal-most extent of that contact a moderately large foramen is present that housed CN III (but not CN VI, contra Weishampel & Bjork, 1989).

Opisthotic/exoccipital

The opisthotics and exoccipitals are indistinguishably fused in this specimen, as is typical for most ornithischians (Galton, 1989), so they are discussed as a single element. The left fused opisthotic/exoccipital is slightly transversely flattened and shifted medially from life positon and is missing the distal end of the paroccipital process. The right fused opisthotic/exoccipital is heavily damaged and the preserved portion is split into multiple pieces separated by matrix filled gaps.

The posteroventral corners of the fused opisthotic/exoccipitals project posteriorly to form the dorsolateral corners of the occipital condyle (Fig. 8B). The ventromedial margins are separated by the dorsal surface of the basioccipital, which forms a small portion of the ventral margin of the foramen magnum. The fused opisthotic/exoccipital forms the majority of the foramen magnum, although transverse crushing has damaged the dorsal margin of the foramen magnum, making it uncertain if the fused opisthotic/exoccipitals contact each other along the dorsal midline or if the supraoccipital formed the dorsal-most portion of the foramen magnum. The posterolateral margin of the fused opisthotic/exoccipital is deeply concave laterally as a result of the posterolateral extension of the paroccipital process (Fig. 8C: pop). The distal ends of both paroccipital processes are not preserved, so the morphology of that structure is unknown.

The ventral margin of the opisthotic/exoccipital makes a firm contact with the dorsolateral margin of the basioccipital, while the anterior margin contacts the prootic with the posterodorsal process of the prootic extending onto the dorsolateral surface of the opisthotic/exoccipital (Fig. 8C). The crista prootica extends slightly onto the dorsolateral portion of the paroccipital process (Fig. 8C: cpr). Along the ventral portion of the contact with the prootic the opisthotic/exoccipital forms the posterior margins of the fenestra ovalis (Fig. 8C: fo) and foramen metoticum (Fig. 8C: fm), as well as the posterior portion of the crista interfenestralis (Fig. 8C: ci). The crista interfenestralis extends posterodorsally onto the lateral surface of the opisthotic/exoccipital as a sharp ridge that divides the grooves extending from the fenestra ovalis (for the stapes) and the foramen metoticum. The groove extending from the fenestra ovalis is bordered dorsally by the crista prootica. The groove from the foramen metoticum is bordered ventrally by another pronounced ridge that extends from the posterodorsal corner of the basal tubera onto the anterolateral surface of the opisthotic/exoccipital, the crista tuberalis (Fig. 8C: ct). The posterior margin of the crista tuberalis is indented by the first in a series of four foramina that pierce the ventral portion of anterolateral surface of the opisthotic/exoccipital. The anterior two foramina were for CN X (vagus nerve) and CN XI (accessory nerve), while the posterior two foramina accommodate two branches of CN XII (hypoglossal nerve).

The dorsal margin of the paroccipital process is flattened to slightly convex and fits against the ventral margin of the medial process of the squamosal. The dorsomedial surface of the opisthotic/exoccipital was broadly overlapped by the supraoccipital (Fig. 8B). The opisthotic/exoccipital did not contribute to the post-temporal foramen, although there may have been a slight groove on the posterodorsal surface leading away from that foramen for the vena capitis dorsalis. No portion of the inner ear canals can be positively identified in this specimen.

Prootic

The prootics form the lateral walls of the braincase and are incompletely preserved on both sides. While the left side is more complete, the preserved portion of the right side is much less distorted. The dorsal margin of the prootic is dorsally concave and fits against the anteroventral margin of the supraoccipital (Fig. 8C). The dorsal half of the lateral surface is dorsolaterally convex. Forming the ventral border of this broadly convex surface is a prominent groove extending roughly anteroposteriorly across the lateral surface, deepening anteriorly. This groove extends just dorsal to the foramen for CN V and touches the contact between the prootic and the laterosphenoid at the spot where the foramen for the vena cerebralis media is located (Fig. 8C). The anterior margin of the prootic has an extensive, slightly sinuous contact with the posterior margin of the laterosphenoid (Fig. 8C). The large foramen for CN V (trigeminal nerve) pierces the anterolateral surface of the prootic, extending to the contact with the laterosphenoid. Posterior to the foramen for CN V an anteroposteriorly thick strut of bone forms the posterior portion of the prootic. Within this strut level with the foramen for CN V is a small foramen that housed CN VII (facialis nerve). A narrow groove extends from the latter foramen ventrally and slightly anteriorly onto the basisphenoid, posterior to the preotic pendant and entering the dorsal margin of the Vidian canal. That groove housed the ramus palatinus (CN VIIIp) of the facialis nerve.

The posterodorsal corner of the prootic extends posterolaterally onto the anterolateral surface of the fused opisthotic/exoccipital. A broad swelling, the crista prootica (Fig. 8C: cpr), is present on this posterodorsal process that forms the dorsal border over two foramina set within a fossa along the posterior margin of the prootic. The dorsal-most foramen is the fenestra ovalis while the ventral foramen is the foramen metoticum. These two foramina are separated by a narrow splint of bone, the crista interfenestralis (Fig. 8C: ci). The stapes (not preserved) inserted into the fenestra ovalis and extended posterolaterally in a groove ventral to the crista prootica that extends from the prootic onto the anterolateral surface of the fused opisthotic/exoccipital. The foramen metoticum is the exit for CN IX (glossopharyngeal nerve) and the vena jugularis interna (Galton, 1989). The ventral margin of the prootic forms a long contact with the basisphenoid, although this area is not well-preserved on either side of the braincase.

Laterosphenoid

Both laterosphenoids are preserved, although the anterodorsal end of the right laterosphenoid is missing. The posterior end is dorsoventrally broad where it makes extensive, slightly sinuous contact with the anterodorsal margin of the prootic dorsal to the foramen for CN V. The posterodorsal corner also makes a slight contact with the supraoccipital. The dorsal margin makes an extensive contact with the parietal. The posterior portion of the dorsal margin is relatively transversely narrow, but moving anteriorly a prominent wing arises along the lateral margin that extends anterolaterally, following the curve of the ventral margin of the parietal. As a result, the anterodorsal margin of the laterosphenoid is mediolaterally wide, making extensive contact with the posteroventral surface of the frontal medially and extending laterally to contact the medial margin of the postorbital along the rounded anterolateral head (Fig. 7B). The anteromedial corner of the laterosphenoid forms the lateral wall of the foramen for CN I (Fig. 8A). The anteroventral margin makes a long, slightly concave contact with the posterior margin of the orbitosphenoid, which is not fused to the laterosphenoid in this specimen (Fig. 8A). A foramen is present between the orbitosphenoid and the laterosphenoid at the ventral-most end of their contact through which CN IV passed. The ventral-most margin of the laterosphenoid is anteroposteriorly straight and contacts the anterodorsal end of the basisphenoid anteriorly and an anterior projection of the prootic posteriorly just dorsal to the foramen for CN V. The laterosphenoid appears to contribute to the anterodorsal margin of the foramen for CN V and a broad groove extends from the anterodorsal corner of that foramen along the contact between the prootic and the basisphenoid that continues anterodorsally onto the posteroventral corner of the laterosphenoid. That groove extends anteriorly to the posterior border of the foramen for CN IV where it then extends anteriorly onto the orbitosphenoid. A sharp ridge overhangs the dorsal margin of that groove on the laterosphenoid. Where that groove first contacts the ventral margin of the laterosphenoid, a foramen is present that indents the ventral margin of the laterosphenoid through which the vena cerebralis media (middle cerebral vein) passed (Fig. 8C: vcm), as in many ornithischians (Galton, 1989). Thus, the groove on the laterosphenoid likely housed the both the ramus ophthalamicus (CN V1) and the vena cerebralis media secunda (Figs. 8A and 8C: gCN V1 & vcms).

Orbitosphenoid

The posterior portions of both orbitosphenoids are preserved in contact with the anteroventral surfaces of the laterosphenoids and the anterodorsal corners of the basisphenoid (Fig. 8A: os). The orbitosphenoids appear to be fused together into a single element, although they remain unfused to the rest of the braincase. In lateral view the orbitosphenoid is ‘crescent-shaped,’ being concave anteriorly and convex posteriorly, while in anterior view the paired orbitosphenoids are ‘oval-shaped,’ being much taller dorsoventrally than they are transversely wide. The dorsomedial margin of the orbitosphenoid forms the ventral margin of the olfactory canal for CN I (olfactory nerve). In anterior view, the ventral half of the orbitosphenoid is dominated by a large foramen for CN II (optic nerve). Dorsal and ventral to the foramen for CN II the orbitosphenoids are damaged, indicating that the anterior portions of the orbitosphenoids were ossified, although not preserved. A deep groove extends dorsolaterally on the preserved anterior surface from the dorsal margin of the CN II foramen. The foramen for CN IV (trochlear nerve) is positioned near the ventral portion of the contact between the orbitosphenoid and the laterosphenoid, a position also seen in the basal ornithopod Thescelosaurus neglectus (Boyd, 2014), the basal iguanodontian Tenontosaurus dossi (Winkler, Murry & Jacobs, 1997); and the dryosaurid Dryosaurus altus (Galton, 1989), but unlike in the hadrosauriform Mantellisaurus atherfieldensis (Norman, 1986) where this foramen is positioned entirely within the orbitosphenoid. A groove extends anteriorly from the anteroventral corner of this foramen onto the lateral surface of the orbitosphenoid that marks the path of the ramus ophthalamicus of the trigeminal nerve (CN V1). Slightly ventral to the foramen for CN IV, a foramen is present along the contact between the basisphenoid and the orbitosphenoid. Weishampel & Bjork (1989) suggested that this foramen housed both CN III (oculomotor nerve) and CN VI (abducens nerve). In Mantellisaurus atherfieldensis CN III and CN VI exit along the contact between the orbitosphenoid and the basisphenoid from two closely situated foramina (Norman, 1986); however, in the basal ornithopods Thescelosaurus neglectus and Thescelosaurus assiniboiensis the foramen for CN VI exits the floor of the braincase and enters the sella turcica (Galton, 1989; Boyd, 2014). Additional preparation work on SDSM 8656 shows that the latter condition was also the case in Dakotadon lakotaensis. Thus, the foramen along the contact between the orbitosphenoid and the basisphenoid only contained CN III.

Supraoccipital

The supraoccipital forms the posterodorsal portion of the braincase and is visible in this specimen in posterior and lateral views (Fig. 8B). It is situated ventral to the parietal and dorsomedial to the fused opisthotic/exoccipitals. The posterior surface of the supraoccipital is anterodorsally inclined and a transversely broad, dorsoventrally oriented ridge is present on the posterior surface, extending from the dorsal margin to near the ventral margin. The dorsal-most peak of the supraoccipital is concave to fit against the slightly convex ventral midline of the parietal while the dorsolateral surfaces are concave to fit against the lateral wings of the parietal. Transverse crushing of this specimen has pushed the fused opisthotic/exoccipitals together along the midline, obliterating the dorsal margin of the foramen magnum and making it impossible to determine with certainty if the supraoccipital contributed to the foramen magnum. The post-temporal foramen indents the lateral margin of the supraoccipital just ventral to the contact with ventromedial corner of the parietal (Fig. 8B: ptf). The supraoccipital forms the medial, lateral, and ventral borders of the post-temporal foramen, while the parietal forms the dorsal margin (contra Weishampel & Bjork (1989) who state that the supraoccipital and the squamosal form the borders). Dorsal to the post-temporal foramen, a dorsolaterally directed projection of the supraoccipital cups the ventromedial corner of the parietal. The posterolateral corner of the supraoccipital makes a small contact with the ventromedial corner of the medial wing of the squamosal just lateral to the post-temporal foramen. The ventrolateral margin of the supraoccipital makes a broad contact with the dorsally concave dorsal margin of the prootic. About halfway along that contact a small foramen is present through which the vena parietalis passed. Immediately dorsal to that contact, the ventral margin of the parietal contacts the dorsolateral surface of the supraoccipital. The anteroventral corner of the supraoccipital makes a small contact with the posterodorsal margin of the laterosphenoid just dorsal to the contact between the latter element and the prootic.

Mandible

Predentary

Weishampel & Bjork (1989) reported that the entire ventromedial process of the predentary is preserved in SDSM 8656; however, the ventromedial process was posteriorly bifurcated, not undivided, and the majority of the left process missing (Fig. 9G). The ventromedial processes overlapped broad facets on the anteroventral surfaces of the dentaries. A pair of large foramina pierce the anterior surface of the predentary, each one slightly offset from the midline (Fig. 9G). Prominent grooves extend ventrolaterally from these foramina to the corner formed by the divergence of the ventromedial and lateral processes. These grooves continue onto the ventral surface of the anterior-most tip of the dentaries and lead to the rostral dentary foramina (sensu Weishampel & Bjork, 1989) (Fig. 9G: rdf). Other than these grooves, the anterior and lateral surfaces of the predentary are largely smooth. The contact surfaces for the dentaries on the lateral processes are oriented ventromedially. The dorsal surfaces of the lateral processes are largely flat except for the lateral margin, along which a thin, rugose ridge is present (Fig. 9H). This ridge is pierced by numerous small foramina that extend from the oral margin ventrolaterally to the lateral surface of the predentary. Three anteriorly projected, roughly conical denticles are present anteriorly along the oral margin (the left denticle is missing and the other two are slightly damaged: Fig. 9H). The central denticle is situated along the midline of the element, and deep, ovoid depressions separate this denticle from the lateral two, giving the dorsal margin of the predentary a ‘w-shaped’ appearance in anterior view. Slight grooves extend ventrally from each of these depressions down to the anterior foramina. These denticles and depressions complement those present on the premaxillae, creating an interlocking contact anteriorly between the upper and lower jaws. Crushing along the midline and loss of bone along the posteroventral surface of the premaxilla makes it impossible to determine if a dorsomedial process was present in this species.

Figure 9 Preserved portions of the lower jaw and close up of maxillary dentition of Dakotadon lakotaensis (SDSM 8656).

(A) photograph of anterior portion of right dentary and the predentary in right lateral view; (B) photograph of same in medial view; (C) photograph of same in dorsal view; (D) photograph of middle portion of left dentary in left lateral view; (E) photograph of same in medial view; (F) photograph of same in dorsal view; (G) photograph of predentary in ventral view; (H) photograph of predentary in dorsal view; (I) photograph of posterodorsal portion of left dentary, anterior portion of surangular, and the posterior portion of the coronoid in left lateral view; (J) photograph of posterodorsal portion of left dentary, anterior portion of surangular, and the posterior portion of the coronoid in medial view; (K) close up view of maxillary teeth 6–8 from middle portion of left maxilla. Abbreviations: adf, anterior dentary fenestra; ant, anterior; asf, accessory surangular foramen; dor, dorsal; lat, lateral; med, medial; post, posterior; vent, ventral. Abbreviations: ant, anterior; ar, accessory ridge; co, coronoid; de, dentary; dor, dorsal; lat, lateral; mc, Meckelian canal; med, medial; mx, maxilla; pd, predentary; post, posterior; pr, primary ridge; rdf, rostral dentary foramen; su, surangular; tp, tooth position; vent, ventral; vlp, ventrolateral process of predentary. Scale bars in (G) and (H) equal 3 cm, all others equal 5 cm.

Dentary

Both dentaries are incomplete, but the right dentary preserves the anterior portion (Figs. 9A–9C). The ventromedial margin of the dentary curves medially as it approaches the anterior end (Fig. 9C), creating a medially projecting shelf on the anterior end of the dentary that formed the dentary symphysis and the articulation facet for the ventromedial processes of the predentary. The dentary symphysis is slightly dorsolaterally inclined. Ventral to the dentary symphysis a groove is present that, when combined with its antimere, would have formed a small fenestra that extended from the anterior-most end of the Meckelian canal and exited between the ventromedial processes of the predentary (Fig. 9B). A similar feature may also be present in Barilium dawsoni (NHMUK 28660: Norman, 2011: Fig. 24) and Hypselospinus fittoni (NHMUK R1834: Norman, 2015: Fig. 44). The articulation facet for the ventromedial processes of the predentary consists of a broad depression on the ventral surface of the anterior-most portion of the dentary. This facet is not visible in lateral view (Fig. 9A). A prominent foramen, the rostral dentary foramen, exists on the ventral surface of the dentary between the articulation facets for the ventromedial and lateral processes of the predentary (Fig. 9G: rdf). A groove extends anteromedially from this groove, connecting with a corresponding groove on the predentary that leads to another prominent foramen on the anterior surface of the predentary. The facet for the lateral process of the predentary is steeply posterodorsally inclined and the articulation surface faces anterolaterally.

In lateral view the preserved dorsal and ventral margins of the dentaries are parallel (Figs. 9A and 9D). A series of foramina are present on the lateral surface of the dentary that are in close proximity to the articulation surfaces for the predentary as well as aligned below the tooth row. An anteroposteriorly oriented ridge arises on the lateral surface of the dentary just ventral to the row of foramina below the tooth row. Posteriorly, this ridge creates a broad, dorsomedially inclined shelf lateral to the tooth row (Fig. 9C). This ridge may eventually connect with the rising coronoid process, but the incomplete preservation of this specimen prevents confirmation. The preserved posterior-most portion of the left dentary (Figs. 9I and 9J) clearly shows that this shelf does not continue posteriorly to form a broad shelf between the tooth row and the rising coronoid process, as the posterior-most tooth position is closely appressed to the coronoid process, unlike the condition seen in Hippodraco scutodens (McDonald et al., 2010).

The incompleteness of the dentaries makes it impossible to determine how many teeth were present in the dentary tooth row. Nine complete alveoli and a tenth partial alveolus are preserved on the left side (Fig. 9F), while twelve complete alveoli are preserved on the right side (Fig. 9C). However, the morphology and position of the dentary tooth row varies between the left and right sides. In the right dentary, the anterior-most alveolus is situated medial to the posterior extent of the articulation surface for the predentary, leaving no room for a diastema between the dentary tooth row and the predentary (Fig. 9C). On the left dentary there is a two centimeter gap between the posterior margin of the articulation surface for the predentary and the anterior end of the dentary tooth row (Fig. 9F). There is some damage to the bone in this area, but only the outer-most bone surface is missing and there is no evidence of alveoli in this space. Although the left dentary is less complete than the right dentary, comparisons between the tooth rows is possible by aligning landmarks preserved on both sides (e.g., the medial articulation facet for the splenial). Based on those comparisons, the anterior-most alveolus on the left dentary corresponds in position to the third alveolus on the right dentary. Thus, two alveoli are missing from the left dentary, the anterior-most alveolus positioned medial to the articulation surface for the predentary and the second alveolus immediately posterior to that articulation surface, indicating the presence of a short diastema on the left side is approximately one tooth position in length. There is no evidence on the left dentary of injury, rehealing, or infection that would explain the lack of tooth positions in the area. These differences are likely the result of a developmental abnormality in the left dentary, or are evidence of individual variation in the anterior extent of the dentary tooth row and presence of a diastema between the tooth row and the predentary in this species. In dorsal view the dentary tooth row is bowed medially along the posterior half, while in lateral view the tooth row is relatively straight with a slight downward angle anteriorly for the first two alveoli on the right side.

A triangular depression is present on the medial surface of the dentary immediately dorsal to the Meckelian canal. This depression begins at the level of the sixth dentary alveolus (right side) and increases in dorsoventral height posteriorly (Fig. 9C). Weishampel & Bjork (1989) identified this depression as an articulation surface for the prearticular. However, the prearticular in basal ankylopollexians does not extend along the medial surface of the dentary to this extent. This facet likely was the contact surface for the splenial, based on position of that bone in other basal ankylopollexians (e.g., Theiophytalia kerri: Brill & Carpenter, 2007). There is also a narrow facet present ventral to the Meckelian canal in this same area (Figs. 9B and 9E), suggesting that the splenial completely covered this canal medially starting at the level of the sixth dentary alveolus and continued posteriorly to an undetermined position (dentaries are not complete enough to make this determination).

The posterodorsal portion of the left dentary is preserved as a separate piece that does not connect to the rest of the preserved portion of the left dentary (Figs. 9I and 9J). This piece was not discussed in Weishampel & Bjork (1989), likely because it was still heavily encased in matrix, which has since been removed. The posterior-most alveolus is present, and it is situated medial to the anterior margin of the rising coronoid process (Fig. 9J: tp). There are portions of two teeth present in that alveolus: the damaged root and crown of the erupted tooth and the dorsal tip of the crown of a still forming replacement tooth. The dorsal surface of the dentary lateral to the tooth row slopes steeply up to the coronoid process, unlike in Hippodraco scutodens where a distinct shelf is present between the tooth row and the rising coronoid process (McDonald et al., 2010). The dorsal portion of the posterior margin of the dentary was not straight as previously reconstructed (e.g., Weishampel & Bjork, 1989; Brill & Carpenter, 2007) and observed in some ankylopollexian taxa (e.g., Fukuisaurus tetoriensis: Kobayashi & Azuma, 2003), but posterodorsally inclined as in Theiophytalia kerri (Brill & Carpenter, 2007) (Fig. 9I). The anterior margin of the dentary portion of the coronoid process is posterodorsally inclined and lacks the rostral expansion present in more derived taxa (e.g., Fukuisaurus tetoriensis: Kobayashi & Azuma, 2003). A shallow depression is present on the lateral surface of the dentary portion of the coronoid process near the contact with the surangular and dorsal to the accessory surangular foramen (Fig. 9I: asf).

The posterior-most portion of the dentary tooth row is situated on a medially projecting ridge, and a wide groove is present along the posteroventral margin of this ridge that demarcates the anterodorsal margin of the inframandibular fossa. A thin, posteriorly projecting sheet of bone forms the lateral portion of this groove and overlaps the medial surface of the surangular just ventral to the coronoid.

Coronoid

The majority of the left coronoid is preserved in articulation with the dentary and surangular (Fig. 9J: co). The overall morphology of the coronoid is similar to that of the basal ornithopod Thescelosaurus neglectus (specimen NCSM 15728: Boyd, 2014), with the lobe like dorsal process and the anteroventrally projecting process situated medial to the dentary tooth row. The coronoid fits into a facet on the medial surface of the dentary portion of the coronoid process, overlapping the medial surfaces of both the dentary and the surangular. The dorsal margin of the coronoid is roughened and extends farther dorsally than the dentary, forming the dorsal tip of the coronoid process (Figs. 9I and 9J). The posterior extent of the coronoid is unknown owing to damage. There is a prominent anteroventrally extending process that extends medial to at least the posterior-most dentary alveolus. Although the full extent of this process is unknown owing to damage, the thickness of the broken end of this process indicates it would have extended farther anteriorly. A similar anterior extension is seen in Iguanodon atherfieldensis (Norman, 2004), although it is apparently absent in Lanzhousaurus magnidens (You, Ji & Li, 2005). It is uncertain if the anterior process of the coronoid of D. lakotaensis was relatively short, as in the basal ornithopods Thescelosaurus neglectus (specimen NCSM 15728: Boyd, 2014) and Hypsilophodon foxii (Galton, 1974), or elongate, as in the basal ornithischian Lesothosaurus diagnosticus (Sereno, 1991) and the basal ornithopod Changchunsaurus parvus (Jin et al., 2010). Although obscured somewhat by crushing, there appears to be an articulation facet covering much of the medial surface of the anterior process. This facet may be for the prearticular, which contacts the ventral extent of the coronoid in Lanzhousaurus magnidens (You, Ji & Li, 2005) and possibly in Theiophytalia kerri (Brill & Carpenter, 2007). Alternatively, the posterodorsal corner of the splenial overlies the anterior process of the coronoid in the basal ornithopods Thescelosaurus neglectus (specimen NCSM 15728: Boyd, 2014) and Changchunsaurus parvus (Jin et al., 2010). Given that neither of these bones are preserved in SDSM 8656, it is uncertain what bone fit into this facet.

Surangular

A portion of the anterodorsal margin of the left surangular is preserved in articulation with portions of the left dentary and coronoid (Figs. 9I and 9J). The anterior portion of the surangular inserts into a groove in the posterior portion of the dentary that is formed by the coronoid process laterally and medially by a posteriorly-projecting sheet of bone that is exposed just ventral to the coronoid. The anterior surface of the surangular is shallowly grooved to fit against a slight ridge present on the posterior margin of the dentary. The dorsomedial surface of the preserved portion of the surangular is also overlapped medially by the coronoid (Fig. 9J). The anterior margin of the accessory surangular foramen (sensu Norman, 2004) is preserved, and a prominent groove extends anterodorsally from this foramen to the suture with the dentary, but does not continue onto the dentary. There is a shallow groove along the anterior margin of the surangular dorsal to the accessory surangular foramen, which is also present in the basal ornithopod Thescelosaurus neglectus (specimen NCSM 15728: Boyd, 2014) and possibly in other iguanodontians, but this feature is often obscured by the posterior portion of the dentary.

Accessory ossifications

Supraorbital

The majority of the left supraorbital is preserved compressed against the prefrontal and frontal. Prior descriptions and interpretive reconstructions of this species identified this piece as the posterior portion of the prefrontal (Weishampel & Bjork, 1989; Brill & Carpenter, 2007; Paul, 2008), but extensive preparation of this specimen revealed its true affinities. The distal end is missing, making it impossible to determine if it spanned the entire length of the orbit or if there was an accessory supraorbital present as occurs in some neornithischian taxa (e.g., Thescelosaurus neglectus; Boyd, 2014). However, there is an anteriorly projecting boss present along the orbital margin of the postorbital that may indicate that at least a cartilaginous connection was present (Fig. 7C). Although the supraorbital is preserved appressed to the dorsal margin of the orbit, this was not the natural position for this element. This position resulted from the medial portion of the prefrontal being crushed medially in relation to the distal end of prefrontal and the remainder of the skull roof, pushing the supraorbital into contact with those elements. The prominent, anteroposteriorly elongate ridge present on the dorsolateral portion of the lacrimal likely denotes the ventral margin of the articulation surface for the supraorbital. A similar structure performs that function in the neornithischian Thescelosaurus neglectus (Boyd, 2014). Thus, it is likely that the supraorbital spanned the contact between the lacrimal and the prefrontal, as it does in heterodontosaurid Heterodontosaurus (Crompton & Charig, 1962), the neornithischians Agilisaurus, Orodromeus, and Thescelosaurus (Peng, 1992; Scheetz, 1999; Boyd, 2014), some ceratopsians (e.g., Archaeoceratops; You & Dodson, 2003), and the iguanodontians Dryosaurus, Mantellisaurus, and Thyeiophytalia kerri (Hooley, 1925; Carpenter, 1994; Brill & Carpenter, 2007).

The anterior articulation facet is ‘oval-shaped,’ with the apex pointing anteriorly (Fig. 7B: sa). A dorsoventrally oriented groove subdivides the articulation facet into two parts, the posterior portion being much smaller than the anterior portion. There is no medial process at the posterior end of this facet, unlike the condition seen in Thescelosaurus neglectus (Boyd, 2014). The posterior projecting shaft appears to have been slightly curved (concave medially and convex laterally), but crushing and fracturing of the bone makes it impossible to determine exactly to what degree (Fig. 7B). The shaft becomes more mediolaterally flattened as it extends posteriorly, with the dorsal and ventral margins forming rounded ridges. The ventral, lateral, and dorsal surfaces are slightly rugose, while the medial surface is smooth and broadly convex.

Dentition

Maxillary dentition

The maxillary crowns are much taller dorsoventrally than they are anteroposteriorly wide (Fig. 9K and Table 1). The complete tooth row is preserved in the left maxilla, displaying the presence of at least nineteen tooth positions, although a twentieth may be present (Fig. 4C). The crown grades gradually onto the root, with only a slight angle present between the two parts. Each tooth position possesses a single replacement tooth that is situated medial to the erupted tooth. A primary ridge is present on the labial surface of the crown that is posteriorly offset (Fig. 9K: pr). Unlike in the dentary crowns, the primary ridge does not display a slight concavity along the midline. Several secondary ridges are present both anterior and posterior to the primary ridge (Fig. 9K: ar). Pronounced denticles are present along the distal margins of the crowns. The apex of the maxillary crown is posteriorly offset, and the posterior margin is steeply sloped while the anterior margin is more gradually inclined. Intercrown spaces are absent between tooth positions, forming a continuous occlusal surface with a single crown participating from each tooth position. The posterior margin slightly overlaps the anterolabial surface of the adjacent crown. The posterior portion of the occlusal surface on each crown is slightly more worn than the anterior portion.

Table 1 Selected measurements of the skull and dentition of Dakotadon lakotaensis (SDSM 8656).

All measurements taken in mm using digital calipers and reported values are the mean values of three independent measurements (raw data not reported). Tooth crown width measurements in parentheses are maximum widths, all other widths were taken at the occlusal surface and represent the preserved maximum width.

Measurement	Height	Width	Length	
Left Maxilla Tooth Position 1	6.15	13.75	–	
Left Maxilla Tooth Position 2	7.04	10.74	–	
Left Maxilla Tooth Position 3	10.53	11.30	–	
Left Maxilla Tooth Position 6	24.20	13.83	–	
Left Maxilla Tooth Position 7	24.8	13.17	–	
Right Maxilla Tooth Position 2	8.73	13.08	–	
Right Maxilla Tooth Position 3	19.19	11.61	–	
Right Maxilla Tooth Position 4	19.72	14.06	–	
Right Maxilla Tooth Position 5	21.93	10.98	–	
Right Maxilla Tooth Position 6	24.61	13.18	–	
Right Maxilla Tooth Position 7	27.21	14.66	–	
Left Dentary Tooth Position 1	23.76	19.03	–	
Left Dentary Tooth Position 2	21.61*	17.06	–	
Left Dentary Tooth Position 3	18.44	21.91	–	
Left Dentary Tooth Position 4	–	(25.52)	–	
Left Dentary Tooth Position 5	21.15	23.35	–	
Left Dentary Tooth Position 6	31.40*	(25.92)	–	
Left Dentary Tooth Position 7	18.62	22.83	–	
Left Dentary Tooth Position 8	36.24	(24.97)	–	
Left Dentary Tooth Position 9	14.14	19.98	–	
Right Dentary Tooth Position 4	24.95	(21.31)	–	
Right Dentary Tooth Position 5	17.26	16.09	–	
Right Dentary Tooth Position 6	22.15	(23.62)	–	
Right Dentary Tooth Position 7	12.43	20.98	–	
Right Dentary Tooth Position 8	29.86	(21.60)	–	
Right Dentary Tooth Position 9	10.27	10.67		
Total Length of Left Maxillary Tooth Row	–	–	204.67	
Total Width of Premaxillae	–	95.85	–	
Total Length of Left Premaxilla	–	–	209.33	
Maximum Dorsoventral Height of Maxilla	101.83	–	–	
Maximum Width of Occipital Condyle	–	61.81	–	
Maximum Height of Left Dentary (Teeth Excluded)	84.14	–	–	
Caudal Vertebra 1	69.46	64.16	80.92	
Caudal Vertebra 2	69.71	64.64	81.27	
Notes.

* Measurement of maximum exposed height, ventral-most extent of crown not exposed.

Dentary dentition

The dentary crowns are typical of basal iguanodontians, with mesiodistally broad, ‘shield-like’ lingual surfaces that grade smoothly onto the root, owing to the lack of a cingulum (Figs. 9B and 9E, Table 1). Enamel is restricted entirely to the lingual surfaces of the dentary crowns. Prominent, smooth denticles are present along the margins of the dentary crowns. These denticles arise posteriorly on a slight, curved shelf situated just ventral to the widest point on the crown and continue along the dorsal margin of the crown, terminating along the anterior margin of the tooth just ventral to the widest point on the crown. The primary ridge on the dentary teeth is posteriorly offset. Secondary ridges are also present on the lingual surfaces, most commonly positioned anterior to the primary ridge. These secondary ridges are often as well-developed as the primary ridges. As noted by Weishampel & Bjork (1989), faint depressions are often present along the midline of both the primary and secondary ridges, giving those ridges a flattened appearance. Occasionally a few faintly-developed subsidiary ridges are also present on the dentary crowns. Intercrown spaces are lacking between the dentary teeth, resulting in the formation of a continuous occlusal surface across the dentary tooth row. Only a single crown from each alveolus contributes to the occlusal surface at a time, and a single replacement tooth is also present in each tooth position. Weishampel & Bjork (1989) noted the presence of two wear facets on many of the dentary crowns, with the posterior facets typically larger than the anterior facets. Many of the worn crowns are chipped and damaged, making it difficult to confirm this observation; however, on some crowns there is a small step along the occlusal surface, and this may be the feature those authors were describing.

Phylogenetic Methods and Results

The phylogenetic analysis presented herein is modified from McDonald (2012). Starting with the character matrix published in that study, fifteen character states were modified for D. lakotaensis based on new information revealed during this study (format = characterstate: 151, 200, 210, 220, 250, 52?, 540, 561, 670, 680, 70?, 720, 77?, 791, and 800). The taxa Bolong yixianensis, Koshisaurus katsuyama, and Proa valdearinnoensis were added to this dataset based on the scorings reported in McDonald et al. (2012: Table 1), Zheng et al. (2014: Appendix 1 (excluding data from immature specimens)), and Shibata & Azuma (2015: Table 2). Changes to this dataset were also made for several taxa based on the suggestions made by Gasca et al. (2015: Appendix 1 (for Delapparentia tulorensis and Barilium dawsoni)) and Shibata & Azuma (2015: Table 3 (Fukuisaurus tetoriensis)). The character codings for the terminal taxa Kukufeldia tilgatensis were combined with Barilium dawsoni, based on the recent synonymization of those two taxa (Norman, 2015). One scoring for Barilium dawsoni (171) was modified (i.e., 170) to reflect the reassignment of NHMUK R1834 from Barilium dawsoni to Hypselospinus cf. fittoni (Norman, 2015). The full dataset of 68 operational taxonomic units (OTUs) was analyzed using TAXEQ3 (Wilkinson, 2001), which identified six OTUs (Callovosaurus leedsi; “Camptosaurus” valdensis; Draconyx loureiroi; Elrhazosaurus nigeriensis; Gilshades ericksoni; and, NHMUK R8676) as taxonomic equivalents that could be removed from the analysis using the principle of safe taxonomic reduction (Wilkinson, 1995). The resulting dataset contained 135 characters for 62 OTUs and was analyzed in the program Tree analysis using New Technology (TNT: Goloboff, Farris & Nixon, 2008). Lesothosaurus diagnosticus was selected as the outgroup and all characters were run unordered (=non-additive in TNT). Branches were collapsed if the minimum length was zero. A traditional search was run for 10,000 replicates, each using a Wagner starting tree with a random seed of 1. The tree bisection and reconnection (TBR) swapping algorithm was used and up to 100,000 trees were saved per replicate. That analysis completely filled the tree buffer with 99,999 most parsimonious trees (MPTs) with a length of 411 steps before the analysis was automatically terminated. The resulting strict consensus topology was almost completely unresolved, except amongst some of the non-ankylopollexian iguanodontian taxa, matching the results obtained by McDonald (2012) from a similarly designed analysis.

Table 2 Results of the series of analyses used to determine which operational taxonomic units (OTUs) should be included in the final phylogenetic analysis presented in Fig. 10.

Numbers in each cell indicate the number of most parsimonious trees (MPTs) obtained during each analysis. Column headings indicate the lengths of the smallest set of MPTs obtained during each round of analyses. All OTUs that were excluded from the final analysis (Fig. 10) are denoted by *.

	314	334	336	336	338	340	342	347	361	362	362	364	366	367	368	370	
Starting 34 OTUs	38																
P. valdearinnoensis	–	19															
L. transoxiana	–	26	13														
C. prestwichii	–	38	19	13													
D. altus	–	38	19	13	13												
K. katsuyama	–	54	19	13	13	13											
D. lettowvorbecki	–	38	19	13	13	13	13										
M. langdoni	–	38	19	13	13	13	13	13									
B. yixianensis	–	75	19	13	13	13	13	13	13								
H. fittoni	–	35	37	21	21	21	21	21	21	8							
O. depressus	–	318	130	91	91	91	91	91	91	13	8						
T. sinensis*	–	40	20	16	16	16	16	16	16	16	11	11					
C. agilis*	–	110	61	39	39	39	39	39	39	39	24	24	15				
K. coetzeei*	–	38	19	13	13	26	26	26	26	26	16	16	22	30			
L. atopus*	–	46	26	90	90	90	90	90	90	90	72	72	99	40	80		
L. arenatus*	–	160	417	264	107	107	107	107	107	39	24	24	33	45	90	240	
H. foulkii*	–	156	79	65	65	65	65	65	65	65	40	40	55	40	80	280	
R. suranareae*	–	38	38	63	63	63	63	63	63	50	29	29	37	63	126	336	
O. hoggii*	–	286	151	97	117	117	117	117	117	78	48	48	66	90	180	480	
P. weishampeli*	–	39	22	28	28	28	28	28	28	38	23	53	56	150	300	800	
P. venenica*	–	180	158	117	117	117	117	117	117	143	88	88	121	165	330	880	
J. meniscus*	–	305	163	162	162	162	162	162	162	162	103	105	143	201	402	1,072	
C. crichtoni*	–	900	480	360	360	360	360	360	360	280	186	186	248	318	636	1,696	
F. tetoriensis*	–	38	38	26	26	26	26	26	26	26	16	184	253	345	690	1,840	
D. turolensis*	–	38	94	65	65	65	65	65	65	26	16	184	252	345	690	1,840	
O. nigeriensis*	–	104	20	89	89	89	89	89	89	36	26	299	367	690	1,380	4,370	
J. yangi*	–	112	29	23	23	23	23	23	23	33	366	1,110	1,155	3,195	6,480	20,022	

Unlike McDonald (2012), we did not proceed to calculate a maximum agreement subtree from that set of MPTs because we have philosophical objections to the methodology behind that practice (i.e., removing OTUs from the final tree topology to seemingly improve resolution even though character data from the full set of included OTUs influenced the final results). Instead, we temporarily trimmed the dataset down to only include those OTUs present in the maximum agreement subtree presented by McDonald (2012: Fig. 1) and added L. diagnosticus (to be used as the outgroup) and the taxon of interest, Dakotadon lakotaensis, leaving 34 OTUs. That dataset was analyzed in TNT using the same methods outlined above, producing 38 MPTs of 314 steps, the strict consensus of which was relatively well-resolved (results not shown). Next, a series of analyses was conducted to determine the effect individually adding each of the 24 previously excluded OTUs had on the recovered number of MPTs (2 OTUs representing individual specimens [NHMUK R3741 and NHMUK R1831] not previously assigned to a described taxon were excluded from consideration). After those analyses were completed, the OTU that most reduced the recovered number of MPTs was retained in the dataset, and another round of analyses was conducted. In situations where multiple taxa produced the same number of MPTs, the OTU that resulted in the fewest steps being added to the lengths of the MPTs was selected. This process continued until the addition of any of the remaining OTUs resulted in an increase in the number of MPTs, although additional analyses were conducted to ensure that the number of MPTs continued to increase as OTUs were added (see Table 2 for results). Via this method ten additional OTUs were added to the final dataset (Table 2), resulting in a final character matrix of 135 characters for 44 OTUs (see McDonald (2012) for character descriptions; see Gasca et al. (2015) for modification of character 110; see Appendix 1 for full character matrix). Analysis of that dataset in TNT was conducted using the same methods detailed above, resulting in the recovery of 8 MPTs of 362 steps. A strict consensus tree was calculated from that set of MPTs and is presented in Fig. 10A. Jackknife support values were calculated for the strict consensus tree using TNT, with the removal probability set at 36% and 1,000 traditional search replicates conducted. The output values were absolute frequencies, and all values over 50% are provided in Fig. 10A situated below their respective nodes.

Figure 10 Systematic relationships of Dakotadon lakotaensis.

(A) strict consensus tree calculated from eight most parsimonious trees obtained from parsimony-based analysis of relationships; (B) clade credibility tree resulting from the posterior probability-based analysis of relationships. In (A), jackknife values over 50% are listed below their respective nodes. In (B), clade support values over 60% are listed below their respective nodes. Taxa with grey branches in (B) are recovered in different positions than in (A).

The final dataset from this study (135 characters for 44 OTUs) was also analyzed using the phylogenetics program MrBayes (v. 3.2.4: Ronquist et al., 2012), which uses the posterior probability optimality criterion rather than the parsimony optimality criterion used by TNT. This was done by exporting the dataset into the appropriate MrBayes format using the program Mesquite (v. 3.0: Maddison & Maddison, 2009) with the default MrBayes block of settings included (e.g., rates = invgamma; burninfrac = 0.25; samplefreq = 1,000; nchains = 4). The file was then opened in MrBayes and a Markov Chain Monte Carlo (MCMC) analysis was run with L. diagnosticus set as the outgroup and all other settings left as the default for standard (i.e., morphological) data. The analysis was run for 25,000,000 generations and the average standard deviation of split frequencies quickly dropped below 0.01. The program Tracer (v. 1.6: Rambaut et al., 2014) was used to assess whether the Effective Sample Size (ESS) value was large enough to ensure that an acceptable number of independent (i.e., uncorrelated) samples was reached, indicating that the analysis was run for a sufficient number of generations and the sample should well represent the posterior distribution. All ESS values were >700, indicating the analysis was run for an acceptable number of generations. The clade credibility tree was then calculated and is presented in Fig. 10B with the clade support values above 60% listed below each node.

Results

The parsimony-based analysis recovers Dakotadon lakotaensis near the base of Ankylopollexia (Fig. 10A), situated below all other non-hadrosauriform ankylopollexian taxa from the Cretaceous (e.g., Iguanacolossus, Hippodraco, and Theiophytalia) and above the non-hadrosauriform ankylopollexian taxa known from the Jurassic (i.e., Camptosaurus, Cumnoria, and Uteodon). This placement differs from that of the Adams consensus tree reported in McDonald et al. (2012), where D. lakotaensis is positioned above Osmakasaurus, Hippodraco and Theiophytalia (as well as other taxa not included in this analysis). Prior studies had also recovered D. Lakotaensis in a close relationship with Iguanacolossus (i.e., McDonald et al., 2012; Shibata & Azuma, 2015), but this study recovers a clade containing Iguanacolossus and Osmakasaurus that is situated well above D. lakotaensis.

Aside from the position of D. lakotaensis, several other interesting relationships are hypothesized in Fig. 10A. A clade containing the taxa Lanzhousaurus, Bolong, and Barilium is recovered positioned just above D. lakotaensis. While no prior studies have recovered all three of these taxa in a clade together, some studies have suggested a possible close relationship between Bolong and Barilium (Shibata & Azuma, 2015: Adams consensus tree) and between Barilium and Lanzhousaurus (Gasca et al., 2015: modified consensus subtree), though those analyses recover all of those taxa positioned higher within Ankylopollexia than they are in this analysis. The taxa Proa and Koshisaurus are recovered as non-hadrosauriform ankylopollexians by this analysis, conflicting with the results of all prior analysis that included one or both of those taxa (e.g., Zheng et al., 2014; Norman, 2015; Shibata & Azuma, 2015). However, two prior studies that did not include Koshisaurus did recover Proa in a polytomy at the base of Hadrosauriformes that leaves open the possibility of Proa being the sister taxon to Hadrosauriformes (McDonald et al., 2012; Gasca et al., 2015), which matches the results of this study (Fig. 10A). The topology in Fig. 10A also provides no support for a clade of ‘iguanodontoids’ as recovered by Norman (2015: Figs. 48 and 50–52).

The topology recovered by the posterior probability-based analysis (Fig. 10B) is less resolved than the parsimony-based topology (Fig. 10A). Most of the non-hadrosauriform ankylopollexian taxa, including D. lakotaensis, are situated in a large polytomy just above the Cumnoria + Uteodon clade and below a smaller polytomy containing Bolong, Barilium, and Hadrosauriformes. Thus, those results neither support nor contradict the hypothesized relationships of D. lakotaensis presented in Fig. 10A and provide little insight into the relationships of that taxon, though some key differences are present between those topologies. The topology in Fig. 10B recovers Bolong and Barilium relatively closer to Hadrosauriformes than in Fig. 10A, and Lanzhousaurus is not positioned with these two taxa. Proa and Koshisaurus are situated in a clade with Iguanodon at the base of Hadrosauriformes, which agrees more with previously published analyses (Zheng et al., 2014; Norman, 2015; Shibata & Azuma, 2015). The recovery of that clade is similar to the group of ‘iguanodontoids’ recovered by Norman (2015), though it contains far fewer taxa in this study.

Discussion

Only two ankylopollexian specimens are known from the Lakota Formation of South Dakota (Fig. 1: A and B). The first of these to be described was USNM 4753, the holotype of Osmakasaurus (=Camptosaurus) depressus(Gilmore, 1909). The second is SDSM 8656, the holotype of Dakotadon (=Iguanodon) lakotaensis (Weishampel & Bjork, 1989). Although the only overlapping material preserved in these two specimens are some caudal vertebrae, the erection of two species from these specimens was considered justified via the referral of each to previously described genera (Camptosaurus and Iguanodon, respectively). Subsequent taxonomic reviews demonstrated that both of those referrals were incorrect (e.g., Paul, 2008; McDonald, 2011) and each is now placed within their own monospecific genus.

As shown in Figs. 10A and 10B, both D. lakotaensis and O. depressus are recovered within Ankylopollexia but outside of Hadrosauriformes. The placement of those taxa in Fig. 10A supports the separation of these two species, but the large polytomy in Fig. 10B provides no resolution as to whether those taxa are conspecific. The only overlapping material preserved in both of these specimens are vertebral centra from the caudal series. In D. lakotaensis, a pair of ridges extend anteroposteriorly along the ventral surface from the anterior chevron articulation facets to the posterior chevron articulation facets (Figs. 11C and 11F). Similar ridges are absent in O. depressus, potentially differentiating these two taxa. The holotype of O. depressus was collected ninety feet above the uncomfortable contact between the Unkpapa Sandstone and the Lakota Formation, in or near the same horizon that yields abundant specimens of Cycadeoides (Darton, 1901; Gilmore, 1909). That horizon is positioned within the Chilson Member of the Lakota Formation (Gott, Wolcott & Bowles, 1974), lower in section than the type locality of D. lakotaensis (see ‘Geologic Settings’). Given these stratigraphic and morphological differences, separation of D. lakotaensis and O. depressus is here retained until such time as more complete material is referred to either or both of these taxa that facilitates more detailed comparisons.

Figure 11 Caudal vertebrae of Dakotadon lakotaensis (SDSM 8656).

(A) photograph of caudal 1 in lateral view; (B) photograph of same in opposite lateral view; (C) photograph of caudal 1 in ventral view; (D) photograph of caudal 2 in lateral view; (E) photograph of same in opposite lateral view; (F) photograph of caudal 2 in ventral view. Abbreviations: lr, lateral ridge; vr, ventral ridge. Scale bars equal 5.0 cm.

In the original description, DiCroce & Carpenter (2001) list the presence of a pair of low ridges on the ventral surface of the caudal vertebral centra that connect the anterior and posterior chevron facets as one of the three autapomorphies of Planicoxa venenica, observed on one of the paratypes. Those authors also noted that the caudal vertebrae of P. venenica differed from those of Dakotadon (=Iguanodon) lakotaensis in that they are slightly amphicoelous and do not display the strong horizontal ridges present on the lateral surfaces observed in the latter taxon (DiCroce & Carpenter, 2001: p. 190). Those observations were based on the previously poorly prepared caudal vertebral centra of SDSM 8656, one of which was still almost entirely encased in sediment. Both caudal centra preserved with SDSM 8656 are now completely cleaned and exposed, providing a better comparison to those of P. venenica (Figs. 11C and 11F). The caudal centra of D. lakotaensis are slightly amphicoelous and the previously noted strong horizontal ridge is only present on one side of one of the caudal vertebrae. Closer examination shows that the strong horizontal ridge is likely an artifact of deformation and distortion of the caudal centrum, as it is present on one side (Fig. 11B) and only a gently rounded surface is present on the opposite side (Fig. 11A). The less distorted caudal centrum shows no evidence of this horizontal ridge on the lateral surface, although the paired ventral ridges are still present (Figs. 11C and 11F). Thus, the overall shape and morphology of the caudal vertebrae is very similar between P. venenica and D. lakotaensis. No other point of comparison is currently possible between P. venenica and D. lakotaensis. It should also be noted that similar ridges are also present on the caudal vertebrae of H. fittoni (Norman, 2015: Fig. 29). It may be that all three of these taxa are closely related, but deciphering their relationships with each other and within Ankylopollexia must wait for the discovery of more complete specimens. Alternatively, this observed distribution could indicate a wider distribution of this feature among ankylopollexians.

Dinosaurian fossils remain poorly known from the Lakota Formation of South Dakota despite the first discovery of fossils well over a century ago. This leads to the perception that this formation is relatively unfossiliferous. Recently, detailed surveys of other Lower Cretaceous strata previously thought to be poor sources of vertebrate fossils (e.g., Cedar Mountain Formation) resulted in the recovery of unexpectedly diverse faunas. Those results combined with recent success in locating microvertebrate localities within the Lakota Formation (e.g., Cifelli, Davis & Sames, 2014) suggest that detailed paleontological surveys of the Lakota Formation could yield a new and significant vertebrate fauna. Such surveys, along with more detailed studies of the stratigraphy of the Lakota Formation within the northern Black Hills, will be crucial for providing insight into this poorly understood period in dinosaurian evolution within North America.

Supplemental Information

Supplemental Information 1 Character codings for the 68 operational taxonomic units (OTUs) used in the various analyses conducted during this study

All OTUs that were excluded from the final analysis shown in Fig. 10 are noted by ∗. See text for further details. Abbreviations: a, polymorphic coding of 3/4; b, polymorphic coding of 4/5.

Click here for additional data file.

We thank Louis Rossow for original discovery and donation of his specimen to SDSM, Russell and LaVon Yuill for identifying and providing access to the type locality of D. lakotaensis, Mindy Householder for extensive preparation and repair of SDSM 8656, and Taylor Vavra and Rachel Jones for assistance with photography. We thank Andrew Farke, Andrew McDonald, Jeff Person, Karen Poole, Hai Xing and José M. Gasca for comments that greatly improved the quality of this manuscript. We also thank the Willi Hennig Society for proving free access to the phylogenetics program TNT.

Institutional Abbreviations

NCSM North Carolina Museum of Natural Sciences, Raleigh, North Carolina, USA

NHMUK Natural History Museum (formerly BMNH, British Museum of Natural History), London, UK

SDSM South Dakota School of Mines and Technology, Rapid City, South Dakota, USA

TLAM.BA Timber Lake and Area Museum (Bill Alley Collection), Timber Lake, South Dakota, USA

USNM United States National Museum, Washington, D.C., USA.

Additional Information and Declarations

Competing Interests

Author Contributions

The authors declare there are no competing interests.

Clint A. Boyd conceived and designed the experiments, performed the experiments, analyzed the data, contributed reagents/materials/analysis tools, wrote the paper, prepared figures and/or tables, reviewed drafts of the paper.

Darrin C. Pagnac conceived and designed the experiments, contributed reagents/materials/analysis tools, wrote the paper, prepared figures and/or tables, reviewed drafts of the paper.

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
