# Peer review of "Insight on the anatomy, systematic relationships, and age of the Early Cretaceous ankylopollexian dinosaur Dakotadon lakotaensis"

_PeerJ, doi:10.7717/peerj.1263_

## Round 0.1 · original submission · Minor Revisions

This is a thorough and beautifully illustrated description of a poorly known dinosaur specimen. The reviewers' comments (and mine) are accordingly brief!

- The reviewers include some helpful comments and requests for clarification. Please address these either via incorporation into the revised manuscript or through rebuttal in your response letter.
- Reviewer 2 provides some suggestions on the phylogenetic analysis that should be incorporated (particularly as related to the Bayesian analysis).
- I note that there was some feedback on the manuscript via the preprint server (https://peerj.com/preprints/1135v1/#feedback) regarding the phylogenetic data matrix - if possible, please address these comments in revision.

EDITOR'S COMMENTS
- The figures are very nicely done.
- Basic measurements for the specimen should be provided. E.g., tooth dimensions; width of occipital condyle; length of tooth row; breadth across premaxillae; etc.

·

Basic reporting

No Comments

Experimental design

No Comments

Validity of the findings

Review by Andrew T. McDonald

This is an excellent and much-needed redescription of Dakotadon, an important iguanodontian taxon. In addition to the new anatomical data and systematic work, the refinement of the holotype’s age and stratigraphic occurrence relative to other Early Cretaceous North American iguanodontians is a crucial advancement. The paper is in need of minor revision.

Abstract

Early Cretaceous needs to be capitalized.

Planicoxa venenica is said to be equivalent in age to Dakotadon lakotaensis. However, if the ostracod and charophyte-based age for the Lakota Formation is correct, then Planicoxa is significantly younger than Dakotadon. The Poison Strip Member of the Cedar Mountain Formation is considered to be Aptian in age (Kirkland et al. 1999).

Introduction

Your list of notable dinosaur discoveries in the Lakota Formation should include the ankylosaur Hoplitosaurus marshi (Lucas 1901, 1902).

Geologic Setting

If the late Valanginian-early Hauterivian age of the lower Fuson Member is accurate, then it predates the Cedar Mountain and Cloverly formations. The lowermost part of the Cedar Mountain Formation, the lower Yellow Cat Member, is considered to be Barremian in age (Kirkland et al. 1999; Kirkland and Madsen 2007; McDonald et al. 2010a; Ludvigson et al. 2010). The Cloverly Formation is considered to be Albian in age (D'Emic and Foreman 2012).

Emended Diagnosis

The “anterior dentary fenestra”, proposed as an autapomorphy of Dakotadon, appears to be present in some other basal styracosternans, including specimens referred to Barilium (NHMUK 28660, holotype of “Kukufeldia” [Norman 2011]) and Hypselospinus (NHMUK R1834 [Norman 2014]). I would be happy to provide pictures, if necessary.

The differential diagnosis needs to be revised to take into account recent taxonomic revisions. I have enumerated the specific issues below; the numbers correspond to the numbers of the characters listed in the differential diagnosis.

1. “Kukufeldia tilgatensis” (McDonald et al. 2010b) is now considered a junior synonym of Barilium dawsoni (Norman 2011), a position with which I agree (personal communication in Norman 2013). The holotype dentary of “Kukufeldia”, NHMUK 28660, is referable to Barilium.

2. The taxon Owenodon hoggii is misspelled here.

4. The statement that the dentary of Barilium is ventrally inflected probably is based upon NHMUK R1834, which was referred to Barilium by McDonald et al. (2010a). However, Norman (2014) demonstrated that NHMUK R1834 is actually referable to Hypselospinus fittoni.

5, 6, and 18. As with statement number 1, these statements need to take into account that the holotype dentary of “Kukufeldia” is referable to Barilium.

Finally, Hypselospinus appears to have ridges on the ventral surfaces of the middle caudal centra (NHMUK R1632 and R1833; see Fig. 29 in Norman 2014), similar to those of Dakotadon and Planicoxa.

Discussion

The same concerns I raised above regarding the supposed concurrence of Dakotadon and Planicoxa apply here.

Table 1

The following taxonomic names are misspelled here: T. sinensis, H. foulkii, R. suranareae, O. hoggii, and P. venenica.

Figure 1

The taxonomic names Tenontosaurus tilletti and Iguanacolossus fortis are misspelled here.

Furthermore, the map should show T. tilletti occurring in southern Oklahoma as well (Langston 1974; Werning 2012). Also, the Tenontosaurus from Utah has not been firmly established to be T. tilletti, and for now probably is best referred to as Tenontosaurus sp.

Figure 10

According to the phylogenetic analysis and definition of Hadrosauridae proposed by Prieto-Márquez (2010), Telmatosaurus is not a hadrosaurid.

Furthermore, this figure should incorporate the recently named clade Hadrosauromorpha, following the definition of Norman (2014).

·

Basic reporting

No comments.

Experimental design

The method of adding taxa into the analysis stepwise is intriguing. I’m not sure I’d stop at the point at which the number of MPTs increases (though I understand why you would—it’s a clear dividing line). However, I think that having more MPTs isn’t a bad thing, as long as it doesn’t lead to completely unresolved consensus trees. A larger number of trees that include more taxa can give us a better understanding of the phylogeny—in particular, which parts of the tree are better resolved than others. It might be worth considering adding in taxa that create a higher number of MPTs, as long as they don’t cause a large number of nodes to collapse in the consensus. Of course, what qualifies as "large" may be somewhat arbitrary. This isn’t necessarily something that needs to be changed for this paper—just something I mused about in considering this method.

As to the Bayesian analysis, the value of the split frequencies is a good first indicator of a run converging, but does not indicate whether there are any problems with autocorrelation due to a lack of mixing. I would recommend using Tracer (Rambaut et al 2014) to evaluate whether the run has converged using the Effective Sample Size (ESS) of the log likelihood. This is simple to check by opening the p file output by MrBayes in Tracer.

Rambaut A, Suchard MA, Xie D & Drummond AJ (2014) Tracer v1.6, Available from http://beast.bio.ed.ac.uk/Tracer

Validity of the findings

There is a small discrepancy that should be cleared up regarding the rostrodorsal process of the maxilla. In the Emended Diagnosis section, (line 227), the manuscript says “8) rostrodorsal process of maxilla present…”. However, in the full decription of the maxilla (beginning at line 413) the authors state “There is only a single, rostroventrally curved process at the anterior end of the maxilla…”. My guess from the figures is that a rostrodorsal process is present, though it isn’t entirely clear. In any case, the diagnosis or description should be revised so that they are in agreement.

Additional comments

I’m happy to see this paper: it’s great that SDSM 8656 has been further prepared, and that its geologic position has been more precisely studied and described. The figures of the specimen are beautiful, and the thorough descriptions, especially of the palate and braincase, will be helpful to anyone studying iguanodontian cranial anatomy.

A minor copy-editing note: in line 635, there is a stray “e” by itself.

·

Basic reporting

No Comments

Experimental design

No Comments

Validity of the findings

No Comments

Additional comments

The authors have provided an overall reevaluation about the biostratigraphy, morphology, and phylogeny of Dakotadon lakotaensis based on the holotype. The osteological description and comparison are very excellent, and the phylogenetic result seems to be convincing. Given that this MS conforms to the standard and policy of PeerJ, I encourage the editor to recommend the MS as accepted.

---

## Round 0.2 · accepted · Accept

Thank you for your thorough attention to the comments from the reviewers and the preprint server.